# Transitioning Heads Conundrum: The Hidden Bottleneck in Long-Tailed Class-Incremental Learning

**Rahul Vigneswaran**  *rahulvigneswaran@gmail.com*
*Indian Institute of Technology Hyderabad, India*

**Hari Chandana Kuchibhotla**  *ai20resch11006@iith.ac.in*
*Indian Institute of Technology Hyderabad, India*

**Vineeth N Balasubramanian**
*Microsoft Research India*  *vineeth.nb@microsoft.com*
*Indian Institute of Technology Hyderabad, India*  *vineeth.nb@cse.iith.ac.in*

**Reviewed on OpenReview:** *https://openreview.net/forum?id=Hb2Jvi5M7X*

## Abstract

Long-Tailed Class-Incremental Learning (LTCIL) faces a fundamental tension: models must sequentially learn new classes while contending with extreme class imbalance, which amplifies catastrophic forgetting. A particularly overlooked phenomenon is the **Transitioning Heads Conundrum**: as replay buffers constrain memory, initially well-represented head classes shrink over time and effectively become tail classes, undermining knowledge retention. Existing approaches fail to address this because they apply knowledge distillation too late, after these transitions have already eroded head-class representations. To overcome this, we introduce **DE**coupling **R**epresentations for **E**arly **K**nowledge distillation (DEREK), which strategically employs **Early Knowledge Distillation** to safeguard head-class knowledge before data constraints manifest. Comprehensive evaluation across 2 LTCIL benchmarks, 12 experimental settings, and 24 baselines, including Long-Tail, Class-Incremental, Few-Shot CIL, and LTCIL methods, shows that DEREK maintains competitive performance across categories, establishing new state-of-the-art results.

## 1 Introduction

Humans are very good at retaining past knowledge while remaining robust to varying frequencies of exposure to different categories that arrive over time. Machine learning models, in contrast, require repeated exposure to retain past knowledge and at the same time are sensitive to frequency of exposure. Unfortunately, real world data are almost always imbalanced in nature. This presents a natural research problem: *how can we train models that learn continually and capture this ability of humans to recall knowledge with limited data while remaining impartial to class frequency?* This problem has found recent interest as Long-Tailed Class Incremental Learning (LTCIL) Liu et al. (2022); Wang et al. (2024); He (2024), and has gained importance with the rise of continual learning methods.

Similar to class-incremental learning (CIL), a key challenge in LTCIL is catastrophic forgetting, where a model loses previously learned knowledge as new classes are introduced. Any effective solution towards solving LTCIL must address this issue while also handling the inherent data imbalance, where the data follows a long tailed distribution, with some classes having significantly more samples than others—referred to as head and tail classes, respectively during a given task. Training on such imbalanced data presents additional challenges, as models tend to become overconfident on head classes while underperforming on tail classes due to data scarcity Vigneswaran et al. (2021); Zhang et al. (2023). To mitigate catastrophic

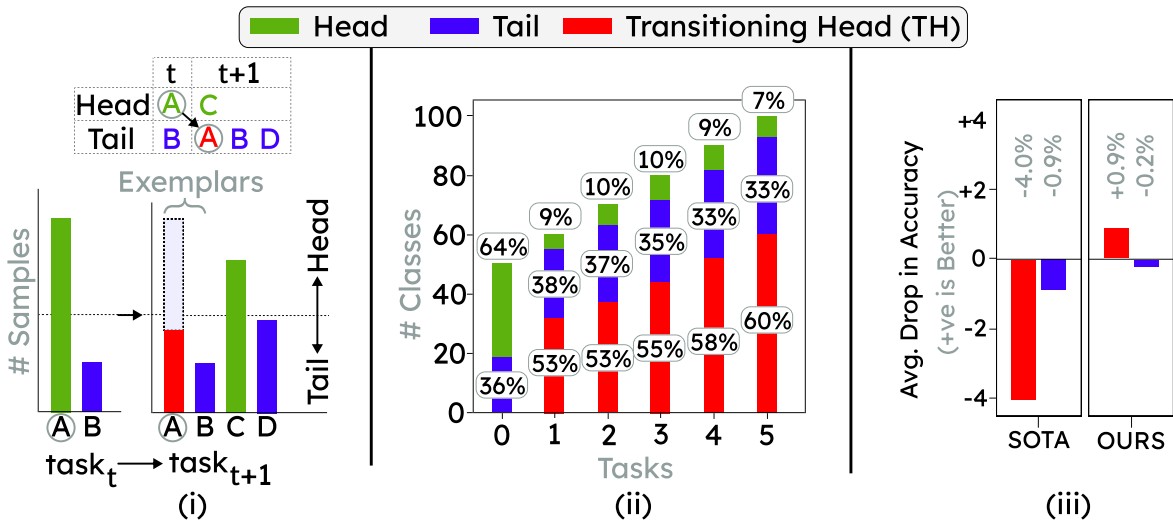

Figure 1: **The Transitioning Head Conundrum.** (i) In LTCIL, a head class $A$ in $task_t$ often becomes a tail class in $task_{t+1}$ due to memory constraints; we term this a **Transitioning Head (TH)**. (ii) Over subsequent tasks, TH samples accumulate, eventually dominating both head and tail distributions. (iii) Unlike SOTA methods, which suffer severe degradation on both TH and tail classes, DEREK effectively preserves TH performance while sustaining tail accuracy.

forgetting, many CIL approaches employ strategies such as storing data samples from previous tasks in a replay buffer Boschini et al. (2022), incorporating regularization constraints Kirkpatrick et al. (2017), and applying logit correction techniques Ahn et al. (2021). Among these, replay-based methods are widely adopted in both CIL and LTCIL Wang et al. (2024); He (2024) as they are simple and are highly effective in reducing catastrophic forgetting.

Our preliminary studies (shown in Fig. 1 (i)) revealed a crucial insight in LTCIL: the limited memory budget in replay buffers leads to head classes (Ⓐ) from **task_t** shrinking in size, eventually becoming tail classes in **task_{t+1}**. Interestingly, since tail classes (Ⓑ) already contain fewer samples than the enforced budget, their entire data is carried over to **task_{t+1}**. We term the classes undergoing this forced change as **Transitioning Head (TH)** classes, as it is always a head class transitioning into a tail and never the other way around. As shown in Fig. 1 (ii), the proportion of TH samples progressively increases across successive tasks, eventually surpassing both head and tail classes.

Going further, we investigated the impact of this issue on model performance in LTCIL. Fig. 1 (iii) shows our observations, wherein we tested against DAKD He (2024) (SOTA in LTCIL) which performs knowledge distillation, a key component of LTCIL and CIL methods. As the figure shows, while distillation performs moderately on tail classes, it experiences a sharp decline in accuracy on TH classes on average across tasks in the LTCIL setting. In contrast, our proposed method DEREK (DEcoupling Representations for Early Knowledge distillation) maintains performance on TH classes while encountering a minor change in performance on tail classes. Understanding and addressing Transitioning Heads Challenge in LTCIL, hitherto unrecognized, forms the key focus of our efforts in this work. Ideally, storing models from each task would preserve the features of TH samples, but this contradicts LTCIL's constraints. A practical alternative is retaining only the previous task's model and applying knowledge distillation. However, knowledge distillation on exemplars is limiting (as shown in Fig. 1 (iii)) and occurs too late when most TH data is already discarded. Complete access to all TH data is only available during **task_t**. Once the model transitions to **task_{t+1}**, most of this data is discarded, making knowledge preservation significantly harder. To address this, we propose performing knowledge distillation at an earlier stage, leveraging the full dataset while it is still accessible rather than relying on a limited set of exemplars. By distilling knowledge at this earlier stage, we hypothesize that the model can learn richer and more accurate representations. Since this distillation occurs before data constraints take effect, we refer to it as **Early Knowledge Distillation**. To address the non-trivial TH

problem and, in turn, provide a solution for LTCIL, we propose DEREK, a method that decouples the learning of Head and Tail classes to enable Early Knowledge Distillation, ensuring better preservation of *Transitioning Head (TH)*, which is crucial to overall performance. While the core principles of DEREK are straightforward, they are meticulously designed to effectively address the challenges posed by TH. Our principal contributions are summarized as follows:

- We identify Transitioning Head (TH) as a critical challenge in LTCIL and demonstrate its significant impact on performance through experiments.

- We propose DEREK, a method designed to enable Early Knowledge Distillation, which we hypothesize is essential for addressing *TH*.

- Our comprehensive experiments demonstrate that DEREK consistently outperforms state-of-the-art approaches across multiple benchmark datasets, and detailed ablation studies further provide deeper insights into its effectiveness.

## 2 Related Work

**Long-Tailed Classification (LT).** The long-tailed classification problem Zhang et al. (2023); Vigneswaran et al. (2021) is a contemporary challenge in deep learning but is often overlooked due to the availability of sterilized, balanced datasets for research. In reality, data naturally follows a long-tailed distribution, adhering to Zipf's law Zipf (1949). This distribution is characterized by a few categories (head classes) containing a large number of samples, while most categories (tail classes) have only a few samples. Such imbalances cause deep learning models to perform well on head classes but poorly on tail classes, resulting in a significant drop in accuracy.

Traditionally, this issue is addressed through re-weighting Cao et al. (2019); Cui et al. (2019); Du et al. (2023) or re-training of the classifier Zhou et al. (2020); Kang et al. (2020). Re-weighting methods assign greater importance to tail classes either at the loss level by inversely weighting their frequencies or by resampling the tail classes more frequently to bias the model towards them. Re-training the classifier involves separating the training of the feature extractor from the classifier. Our method is closest perhaps to GLMC Du et al. (2023) architecture-wise, but the objectives are fundamentally different. GLMC prioritizes consistency across hard augmentations, a focus that may limit its effectiveness in LTCIL setups, where adapting to evolving class distributions is essential. Notably, LT methods aren't designed for continual learning, especially in addressing TH. In contrast, our approach is explicitly tailored to tackle TH in LTCIL.

**Class-Incremental Learning (CIL).** These methods Zhou et al. (2024) aim to learn from a continuous stream of data without catastrophically forgetting previously learned classes. Various strategies have been proposed over the years, including replay-based methods, regularization methods and logit correction methods. Replay-based methods Rebuffi et al. (2017); Boschini et al. (2022); Sarfraz et al. (2023) address forgetting by retraining model with stored data from old tasks alongside new data. Regularization methods Kirkpatrick et al. (2017); Li & Hoiem (2017); Zhu et al. (2021a) tackle this issue by adding constraints that ensure model retains knowledge of old tasks while learning new ones. Additionally, logit correction methods Belouadah & Popescu (2019); Wu et al. (2019); Zhao et al. (2020); Ahn et al. (2021) adjust output scores to prevent bias toward new tasks, aiding the model in retaining past knowledge. Methods like CoSCL Wang et Cl. (2022b) may be similar in spirit to our method, although it focuses only on the CIL task. CosCL utilizes small parallel learners to reduce task interference. While it promotes cooperation among these learners in feature space, learners are not specialized experts and lack explicit mechanisms to support head or tail classes. More critically, their design does not account for TH, as this challenge does not arise in CIL. In contrast, DEREK cultivates specialized expertise within individual learners and facilitates effective knowledge distillation across tasks using novel collaborative loss, essential for addressing TH.

**Long-Tailed Class-Incremental Learning (LTCIL).** This setting brings together Long-Tailed (LT) distributions and the Class-Incremental Learning (CIL) paradigm. In LTCIL, each task within the CIL framework follows a long-tailed distribution, presenting challenges that conventional LT and CIL methods, when applied in isolation, cannot individually address. Notably, all existing approaches to LTCIL, just

like our proposed method DEREK, are two-staged, except for DAKD He (2024), which employs gradient reweighting and dynamically adjusts the intensity of knowledge distillation. LWS Liu et al. (2022) focuses on dynamically scaling classifier weights to improve balance across classes. Subproto Wang et al. (2024) introduces two complementary spaces: sub-prototype space, strengthening class representations by capturing insights from different classes, & reminiscence space, aiding in preserving model's past knowledge. Together, these enhance data while mitigating memory loss. However, distillation in these methods remains limited, as they rely on small set of exemplars and fail to address TH. Moreover, their tightly coupled head and tail representations hinder early knowledge distillation. In contrast, our method enables Early Knowledge Distillation by decoupling these representations and leveraging all available data. Empirical results validate the effectiveness of this approach.

## 3 DEREK: Proposed Methodology

**Problem Definition.** In Class-Incremental Learning (CIL), a model sequentially learns $T$ distinct tasks, each containing a disjoint set of classes. Formally, let $(x_{cls}^{(t)}, y_{cls}^{(t)})$ denote a data sample from class $cls$ in task $t$. A common assumption in standard CIL settings is that each task contains an equal number of samples per class, i.e., $|cls^{(t)}| = N$, $\forall t$. However, Long-Tailed Class-Incremental Learning (LTCIL) challenges this unrealistic constraint by allowing tasks to exhibit a long-tailed class distribution, a detailed description of which is shown in Sec. A.4.

**DEREK: Conceptual Framework.** The significance of providing special focus on Transitioning Head (TH) has already been established in Section 1. In this section, we delve into rationale of proposed method, **DE**coupling **R**epresentations for **E**arly **K**nowledge distillation (DEREK). While each component of DEREK is straightforward in isolation, they are carefully designed to collectively tackle challenges associated with TH effectively. As illustrated in Fig. 1 (iii), Knowledge Distillation alone is insufficient for effectively supporting THs, as its effectiveness is limited by exemplars retained during distillation process when transitioning from $task_t$ to $task_{t+1}$. The only stage in training where full access to all of TH's data is available is during $task_t$, before progressing to $task_{t+1}$. Once model transitions to next task, most of TH's data is discarded, making knowledge retention significantly more challenging. To address this, we propose distilling knowledge earlier, allowing model to learn richer, more precise representations. Since this distillation occurs before data constraints take effect, we term it *Early Knowledge Distillation.*

The key challenge lies in identifying the appropriate teacher and student for the distillation process. DEREK addresses this by leveraging a crucial insight from the learning process. As illustrated in Fig. 1 (i), we deterministically know that a Head class (Ⓐ) in $task_t$ will transition into a Tail class

---

**Algorithm 1** DEREK

**Require:** Tasks: $\mathcal{T}$, Epochs: $e_1, e_2$, Dataset: $\mathcal{D}$,
    Sampling: $\mathcal{S}_{\text{imb}}, \mathcal{S}_{\text{bal}}$, Counts: $n_{\text{head}}, n_{\text{tail}}$,
    Learning rate: $\eta$, $f$: Feature Extractor,
    $g$: Linear Layer, $C$: Classifier, $\sigma$: Softmax

1   **for** *task $t \in \mathcal{T}$* **do**
2     // Section 3.1
3     **for** *epoch $\leftarrow 1$* **to** $e_1$ **do**
4       Sample batch $\mathcal{B} \sim \mathcal{S}_{\text{imb}}(\mathcal{D}_t)$
5       **for** *each $(x, y) \in \mathcal{B}$* **do**
6         $z_{\text{head}} \leftarrow C^{\text{head}}\left(\sum_{j=1}^{n_{\text{head}}} g_j^{\text{head}}(f_j^{\text{head}}(x))\right)$
7         $z_{\text{tail}} \leftarrow C^{\text{tail}}\left(\sum_{k=1}^{n_{\text{tail}}} g_k^{\text{tail}}(f_k^{\text{tail}}(x))\right)$
8         $\mathcal{L} \leftarrow \mathcal{L}_{\text{head}}(z_{\text{head}}, y) + \mathcal{L}_{\text{tail}}(z_{\text{tail}}, y)$
9         // $\theta_1$: params of $f, g, C$ (head/tail)
10        $\theta_1 \leftarrow \theta_1 - \eta \nabla_{\theta_1} \mathcal{L}$
11       **end**
12     **end**
13     // Section 3.2
14     // Freeze: $f_j^{\text{head}}, f_k^{\text{tail}}, g_j^{\text{head}}, g_k^{\text{tail}}$
15     **for** *epoch $\leftarrow 1$* **to** $e_2$ **do**
16       Sample batch $\mathcal{B} \sim \mathcal{S}_{\text{bal}}(\mathcal{D}_t)$
17       **for** *each $(x, y) \in \mathcal{B}$* **do**
18         $z_{\text{head}} \leftarrow C^{\text{head}}\left(\sum_{j=1}^{n_{\text{head}}} g_j^{\text{head}}(f_j^{\text{head}}(x))\right)$
19         $z_{\text{tail}} \leftarrow C^{\text{tail}}\left(\sum_{k=1}^{n_{\text{tail}}} g_k^{\text{tail}}(f_k^{\text{tail}}(x))\right)$
20         $\mathcal{L} \leftarrow \mathcal{L}_{\text{CL}}(\sigma(z_{\text{head}}), \sigma(z_{\text{tail}}))$
21         // $\theta_2$: params of $C^{\text{head}}, C^{\text{tail}}$
22        $\theta_2 \leftarrow \theta_2 - \eta \nabla_{\theta_2} \mathcal{L}$
23       **end**
24     **end**
25   **end**

---

in $task_{t+1}$, thereby explicitly defining the transition path of the TH. To effectively model this transition, DEREK trains specialized experts for head and tail classes, ensuring their representations remain decoupled.

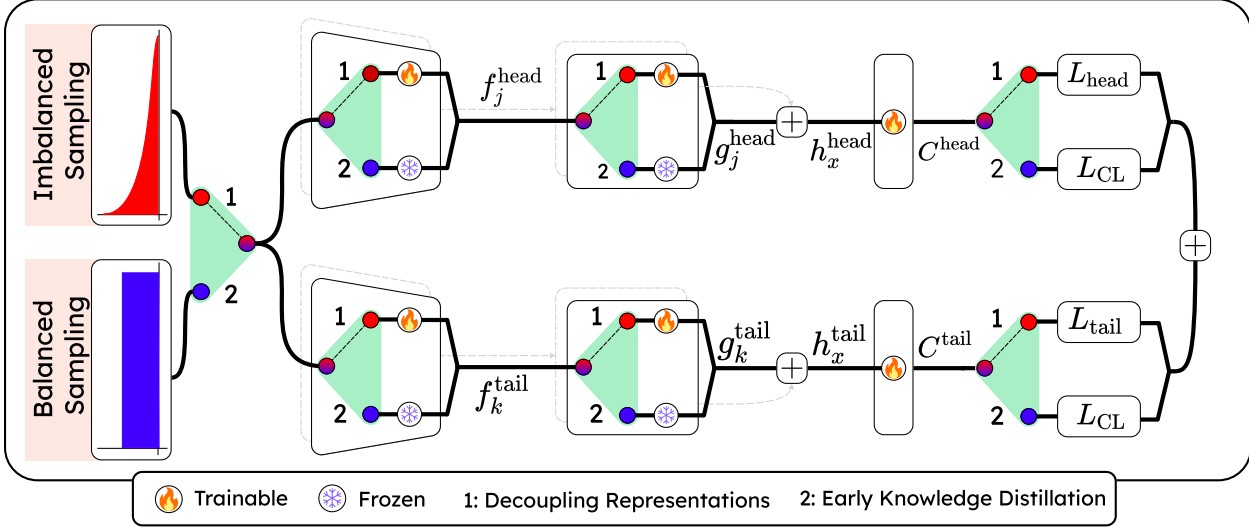

Figure 2: Overview of the proposed method, DEREK, for LTCIL: A switch mechanism is employed for clarity. To decouple the representation (switch at position 1), specialized expertise for head and tail classes is developed. To do an early knowledge distillation (switch at position 2), collaborative loss ($L_{CL}$) is used. Trainable components are marked with a flame icon, and frozen ones with a snowflake.

This results in one expert proficient in head classes (Ⓐ) and another specialized in tail classes (Ⓑ). By integrating our prior observation of TH's exact transition path with these decoupled experts, we establish a logical connection: the head class (Ⓐ), thoroughly learned by the head expert in $task_t$, is subsequently re-learned by the tail expert in $task_{t+1}$ as it transitions into a tail class. Since this transition is known at $task_t$ while we still have full access to the head class data, we can designate the head expert as the teacher and the tail expert as the student. This enables us to perform *Early Knowledge Distillation* before data constraints take effect, ensuring better knowledge transfer and retention. We note that prior methods (He (2024); Wang et al. (2024)) do not exploit this early distillation window because three structural questions remain unresolved in their frameworks: which classes will transition, who should serve as the teacher, and when the distillation window opens. By decoupling representations, DEREK creates the teacher–student pair, and the deterministic nature of the transition defines the distillation window, making early distillation both well-defined and practically applicable.

## 3.1 Decoupling Representations

The objective is to decouple the learning of head and tail classes and improve their performance by addressing their distinct characteristics. Our method effectively addresses this by the use of independent head and tail-aware experts, as illustrated in Figure 2, comprising of $n_{head}$ experts for the head classes and $n_{tail}$ experts for the tail classes. For this, all switches are set to position 1 in Figure 2. Each sample $x$ is passed through all the experts. For the $j$-th head expert, the feature extractor is denoted as $f_j^{\text{head}}(x)$, where $j \in \{1, \ldots, n_{head}\}$, and for the $k$-th tail expert, it is $f_k^{\text{tail}}(x)$, where $k \in \{1, \ldots, n_{tail}\}$. Given the impracticality of heuristically weighting each expert's features, we employ an end-to-end approach to weight all experts using head and tail-specific linear layers, $g_j^{\text{head}}$ and $g_k^{\text{tail}}$, as shown:

$$h_x^{\text{head}} = \sum_{j=1}^{n_{head}} g_j^{\text{head}} \left( f_j^{\text{head}}(x) \right), \quad h_x^{\text{tail}} = \sum_{k=1}^{n_{tail}} g_k^{\text{tail}} \left( f_k^{\text{tail}}(x) \right) \tag{1}$$

where $h_x^{\text{head}}$ and $h_x^{\text{tail}}$ represent the pre-logit activations of the data point $x$. These weighted experts are then classified using their dedicated classifiers $C^{\text{head}}$ and $C^{\text{tail}}$ as follows to obtain the corresponding logits.

$$z_x^{\text{head}} = C^{\text{head}} \left( h_x^{\text{head}} \right), \quad z_x^{\text{tail}} = C^{\text{tail}} \left( h_x^{\text{tail}} \right) \tag{2}$$

Until now, there has been no explicit learning signal guiding head experts to specialize in head classes and tail experts to focus on tail classes. To address this, we introduce specialized loss functions tailored to each expert. For head experts, we adopt the standard cross-entropy loss, denoted as $L_{\text{head}}$, since the natural class imbalance in the data inherently biases learning toward head classes. In contrast, for tail experts, we employ the Class Balanced (CB) loss Cui et al. (2019), denoted as $L_{\text{tail}}$. These losses are formulated as follows:

$$L_{\text{head}} = -\sum_{cls} y \log\left(p_{x,cls}^{\text{head}}\right), L_{\text{tail}} = -\sum_{cls} \left(\frac{1-\beta}{1-\beta^{N_{cls}}}\right) y \log\left(p_{x,cls}^{\text{tail}}\right) \tag{3}$$

where $p_{x,cls}^{\text{head}}$ and $p_{x,cls}^{\text{tail}}$ represent the post-softmax probabilities of the $z_{x,cls}^{\text{head}}$ and $z_{x,cls}^{\text{tail}}$, respectively. Here, $y$ denotes the true label, $\beta$ is a parameter that adjusts the effective number of samples (commonly set to 0.9999), and $N_{cls}$ is the number of training samples for class $cls$ to which the sample $x$ belongs. While CB loss encourages better representation learning for tail classes, it does not inherently address TH forgetting on its own, which remains the primary challenge in LTCIL. Therefore, our approach focuses on explicitly mitigating TH while leveraging CB loss as a supportive component. It is important to note that the issue of TH in LTCIL is distinct from conventional long-tailed classification, where the primary challenge lies in addressing class imbalance. The $L_{\text{head}}$ and $L_{\text{tail}}$ loss functions are designed to ensure that head experts effectively concentrate on head classes while tail experts are better equipped to manage tail classes, thereby fostering complementary expertise.

To illustrate, consider the class *Lamp*, which is a head class in task $t$ with 500 samples under $\rho = 100$. During Stage 1, both experts are trained simultaneously on task $t$'s data. The head expert is trained with $L_{\text{head}}$ (standard CE) and learns a rich representation of *Lamp* from all 500 samples, biased naturally toward high-frequency classes. The tail expert is trained with $L_{\text{tail}}$ (CB loss) and learns representations better suited to lower-frequency categories. By the end of Stage 1, the head expert holds a strong, data-rich representation of *Lamp* while the tail expert is better calibrated for the tail regime that *Lamp* is about to enter.

## 3.2 Early Knowledge Distillation

With the representations of head and tail classes now decoupled, and leveraging the key insight that a head class, currently well-learned by the head expert, will be managed by the tail expert in the next task, we can strategically designate the head expert as the teacher and the tail expert as the source for Early Knowledge Distillation. To achieve this, all switches are set to position 2 in Figure 2. DEREK introduces a novel Collaborative Loss ($L_{\text{CL}}$) to facilitate Early Knowledge Distillation from the head expert to the tail expert. $L_{\text{CL}}$ is formulated as follows:

$$L_{\text{CL}} = \sum_{cls} \left(\left(p_{x,cls}^{\text{tail}} - p_{x,cls}^{\text{head}}\right)^2 - 1\right) \log\left(p_{x,cls}^{\text{tail}}\right), \tag{4}$$

where $p_{x,cls}^{\text{head}}$ and $p_{x,cls}^{\text{tail}}$ represent the post-softmax logits of head and tail experts, respectively. By optimizing $L_{\text{CL}}$, the tail expert gradually refines its predictions to align with those of the head expert, facilitating a smooth transition in handling the TH. When $p_{x,cls}^{\text{tail}}$ is high, $\log\left(p_{x,cls}^{\text{tail}}\right)$ approaches 0. However, if there is significant misalignment between the experts (i.e., $p_{x,cls}^{\text{head}} - p_{x,cls}^{\text{tail}} \neq 0$), the loss remains high, compelling the tail expert to correct its predictions. Furthermore, $L_{\text{CL}}$ exhibits an emergent cross-entropy property: as training progresses and the predictions of both experts become increasingly aligned, the loss naturally converges to a cross-entropy form. This ensures that optimization proceeds as if both experts were learning in unison, maintaining stability and consistency in knowledge transfer. This convergence is particularly well-suited to the long-tail setting. Prior work has shown that fine-tuning classifiers with balanced sampling can correct the representational bias introduced by imbalanced training Kang et al. (2020). The emergent CE property of $L_{CL}$ ensures that once knowledge transfer from the head expert is complete, the tail classifier naturally transitions into this balanced fine-tuning regime. As a result, distillation and classifier correction are combined within a single objective.

For Early Knowledge Distillation, we use the same data as in the decoupling phase but apply balanced sampling instead, as it empirically yields better results (see Section 5). To preserve the specialized knowledge learned by each expert, we freeze the backbones $f_j^{\text{head}}, f_k^{\text{tail}}, g_j^{\text{head}}, g_k^{\text{tail}}$ and fine-tune only the classifiers $C^{\text{head}}$

and $C^{\text{tail}}$, ensuring that the distillation process focuses solely on knowledge transfer without overwriting the existing expert representations.

Continuing the *Lamp* example: at the end of task $t$, we know deterministically that *Lamp* will become a tail class in task $t + 1$, where only a small number of exemplars will be retained. The head expert, having seen all 500 samples of *Lamp*, is designated as the teacher and the tail expert as the student. $L_{\text{CL}}$ transfers this knowledge before the data disappears. When task $t+1$ begins and *Lamp* is now managed by the tail expert, the transferred knowledge allows it to handle *Lamp* far more effectively than if it had encountered it for the first time with only a handful of exemplars.

Algorithm 1 provides a detailed overview of the complete procedure. During inference, logits are computed by both the experts and then averaged with equal weights, ensuring that the final prediction benefits from the collective expertise, leading to more robust outcomes.

## 4 Experiments

### 4.1 Experimental Setup

**Evaluation Protocol & Datasets.** Our study adheres to established protocols from significant prior works in LTCIL, specifically those proposed by Liu et al. (2022); Wang et al. (2024). The evaluation starts with training on the first half of the classes, followed by evenly distributing the remaining classes across $T$ tasks. For example, with 100 classes, a 5-task setup trains on 50 classes first, then 10 classes per task, while a 10-task setup uses 5 classes per task. We employ the CIFAR100-LT Krizhevsky & Hinton (2009) and ImageNet-Subset Russakovsky et al. (2015) datasets, both consisting of 100 classes and adjusted to exhibit long-tailed distributions characterized by an imbalance factor $\rho = \frac{N_{\max}}{N_{\min}}$, where $N_{\max}$ denotes the maximum frequency of a class and $N_{\min}$ denotes minimum frequency of a class. We evaluate across various levels of class imbalance, specifically testing $\rho = 100, 50, 10$ as detailed in Wang et al. (2024). When $\rho = 100$, for CIFAR100-LT the most frequent class has 500 samples, and the least frequent class has 5 samples. For the ImageNet-Subset class, frequency ranges from 1,300 samples for the most frequent class to 13 samples for the least frequent class. To ensure a comprehensive assessment, each strategy is tested under both shuffled and ordered configurations of class sequences Liu et al. (2022), allowing us to critically examine the impact of class sequencing on learning dynamics and overall model performance. All the methods compared against are rehearsal-based.

**Implementation Details.** Following the already established protocol Liu et al. (2022); Wang et al. (2024); He (2024), we employ ResNet32 for CIFAR100-LT and ResNet18 for ImageNet-Subset. For all datasets the model undergoes initial warm-up for $e_0$ epochs with an adaptive learning rate starting at 0.1 (refer Algorithm 2). After this phase, the model is duplicated to create the required number of experts. The model then undergoes decoupling of head and tail representations for $e_1$ epochs, designed to address the specific needs of head and tail classes. The early knowledge distillation follows, involving training the final classifier for $e_2$ epochs using a balanced data sampler to optimize model performance. Notably, both decoupling and early knowledge distillation utilize the same data; however, decoupling employs an imbalanced sampler, while early knowledge distillation uses a balanced sampler. Following Liu et al. (2022), we also freeze the classifier heads of previous tasks to preserve the integrity of learned representations and support our method with Knowledge Distillation. In certain setups, when the performance in the initial stage surpasses that of the subsequent stage, we select the best-performing model from the earlier stage. To stabilize the training, we also apply collaborative loss in both directions.

For CIFAR100-LT, $e_0 = 500$ with learning rate reductions at the 250th, 350th, and 450th epochs. For ImageNet-Subset, $e_0 = 100$ with adjustments at the 30th and 60th epochs. Both $e_1$ and $e_2$ are set to 100 for all datasets. Batch sizes are set at 128 for CIFAR100-LT and 64 for the ImageNet-Subset. All experiments were conducted using a single NVIDIA Tesla V100 GPU equipped with 32GB of memory. In all experiments, we use 20 exemplars per class across tasks.

For further details on comparison methods, evaluation metrics, computational complexity, and the warm-up algorithm, please refer to Sec. A.1.

|  | | Ordered | | | | Shuffled | | | |
| --- | --- | --- | --- | --- | --- | --- | --- | --- | --- |
|  | | CIFAR | | ImageNet | | CIFAR | | ImageNet | |
|  | Method | 5 | 10 | 5 | 10 | 5 | 10 | 5 | 10 |
| **LT** | LDAM Cao et al. (2019) | 20.1 | 19.4 | 19.7 | 18.6 | 11.5 | 11.4 | 15.4 | 15.2 |
|  | BalPoE Aimar et al. (2023) | 24.9 | 24.0 | 23.5 | 23.0 | 18.9 | 17.6 | 17.8 | 17.3 |
|  | MDCS Zhao et al. (2023) | 24.3 | 23.7 | 23.1 | 22.8 | 18.2 | 16.3 | 16.9 | 16.2 |
| **CIL** | iCaRL Rebuffi et al. (2017) | 36.4 | 36.2 | 43.6 | 42.7 | 31.5 | 30.5 | 35.4 | 34.6 |
|  | EWC Kirkpatrick et al. (2017) | 32.1 | 31.6 | 35.7 | 31.2 | 28.7 | 25.3 | 30.8 | 30.4 |
|  | LwF Li & Hoiem (2017) | 32.8 | 31.9 | 36.1 | 33.4 | 29.3 | 25.1 | 31.6 | 31.0 |
|  | IL2M Belouadah & Popescu (2019) | 39.8 | 38.8 | 46.9 | 44.3 | 34.9 | 33.4 | 40.7 | 39.0 |
|  | BiC Wu et al. (2019) | 34.5 | 31.0 | 47.7 | 41.0 | 25.6 | 25.9 | 33.1 | 29.2 |
|  | WA Zhao et al. (2020) | 35.6 | 33.0 | 48.2 | 42.2 | 32.0 | 26.8 | 32.6 | 28.1 |
|  | SDC Yu et al. (2020) | 34.9 | 34.5 | 43.2 | 42.0 | 32.7 | 29.6 | 33.9 | 33.4 |
|  | SSIL Ahn et al. (2021) | 33.9 | 23.1 | 40.0 | 41.7 | 30.7 | 29.2 | 38.9 | 35.1 |
|  | PASS Zhu et al. (2021b) | 35.8 | 35.2 | 43.9 | 42.6 | 33.6 | 31.8 | 34.2 | 33.8 |
|  | IL2A Zhu et al. (2021a) | 40.6 | 40.9 | 47.2 | 46.9 | 35.1 | 36.2 | 40.5 | 39.2 |
|  | FOSTER Wang et al. (2022a) | 38.4 | 35.1 | 47.3 | 46.8 | 37.2 | 37.9 | 46.5 | 43.8 |
|  | TwF Boschini et al. (2022) | 40.1 | 39.8 | 46.1 | 45.4 | 34.2 | 33.8 | 38.6 | 38.1 |
|  | MAFDRC Chen & Chang (2023) | 42.1 | 41.9 | 48.6 | 47.1 | 37.9 | 38.5 | 48.2 | 44.1 |
|  | SCoMMER Sarfraz et al. (2023) | 41.2 | 41.0 | 47.0 | 46.5 | 35.0 | 35.2 | 39.3 | 38.9 |
| **FSCIL** | SAVC Song et al. (2023) | 36.1 | 35.8 | 45.5 | 45.0 | 34.4 | 32.3 | 35.3 | 34.9 |
| **LTCIL** | EEIL+LWS Liu et al. (2022) | 39.9 | 38.8 | 50.9 | 48.3 | 36.4 | 34.9 | 43.6 | 41.4 |
|  | FOSTER+LWS He (2024) | 43.2 | 44.1 | 54.2 | 53.8 | 40.2 | 39.4 | 48.8 | 46.9 |
|  | LUCIR+LWS He (2024) | 45.1 | 43.3 | 54.8 | 53.3 | 38.4 | 37.8 | 48.9 | 47.3 |
|  | PODNET+LWS He (2024) | 44.4 | 43.5 | 54.7 | 54.2 | 38.3 | 38.4 | 48.0 | 47.7 |
|  | DAKD He (2024) | 44.8 | 44.1 | 55.6 | 53.9 | 40.1 | 39.1 | 50.5 | 49.1 |
|  | SubProto Wang et al. (2024) | 44.8 | 44.3 | 52.7 | 52.6 | 40.2 | 39.4 | 45.3 | 44.8 |
|  | DEREK (Ours) | **49.6** | **49.9** | **60.2** | **59.0** | **44.5** | **45.4** | **53.3** | **53.8** |
|  | | 4.8↑ | 5.6↑ | 4.6↑ | 4.8↑ | 4.3↑ | 6.0↑ | 2.8↑ | 4.7↑ |

Table 1: **Imbalance Ratio = 100.** Accuracies by category (LT, CIL, Few-Shot CIL, LTCIL) for 5-task and 10-task setups under ordered and shuffled conditions. **Best** and second-best scores are highlighted, with improvements marked by ↑. DEREK consistently outperforms the baselines.

## 4.2 Results and Discussions

The experimental results for both ordered and shuffled configurations are detailed in Table 1 for $\rho = 100$, Table 2 for $\rho = 50$ and Table 3 for $\rho = 10$. An imbalance ratio $\rho = 100$ represents a highly imbalanced scenario, while $\rho = 10$ is closer to conventional scenarios with less imbalance, providing a more typical distribution of class frequencies.

**Ordered LTCIL.** The results unequivocally demonstrate the superior performance of our proposed method, DEREK, in the ordered LTCIL setting. Traditional LT methods exhibit considerably lower performance, underscoring their limitations in effectively managing the ordered LTCIL scenario. In contrast, CIL methods, although performing better than traditional LT methods, still fall short in maximizing performance. DEREK, however, consistently outperforms state-of-the-art methods such as DAKD He (2024) and SubProto Wang et al. (2024), indicating its robustness and efficacy. On CIFAR100LT dataset, DEREK achieves an average improvement of +5.2% for Imbalance $\rho = 100$ and $\rho = 50$, and +2.6% for Imbalance $\rho = 10$, across 5 and 10 task settings. The improvements are even more pronounced on the ImageNet-Subset dataset, with average gains of +4.7% for Imbalance $\rho = 100$, +7.25% for Imbalance $\rho = 50$, and +10.3% for Imbalance

|  |  | Ordered | | | | Shuffled | | | |
| --- | --- | --- | --- | --- | --- | --- | --- | --- | --- |
|  |  | CIFAR | | ImageNet | | CIFAR | | ImageNet | |
|  | Method | 5 | 10 | 5 | 10 | 5 | 10 | 5 | 10 |
| LT | LDAM Cao et al. (2019) | 24.8 | 23.1 | 27.3 | 26.4 | 13.6 | 8.1 | 19.4 | 19.0 |
|  | BalPoE Aimar et al. (2023) | 28.3 | 27.5 | 30.4 | 28.5 | 20.5 | 20.0 | 22.6 | 21.4 |
|  | MDCS Zhao et al. (2023) | 27.9 | 27.1 | 30.1 | 28.3 | 19.5 | 19.1 | 22.4 | 21.4 |
| CIL | iCaRL Rebuffi et al. (2017) | 40.2 | 39.4 | 47.9 | 47.5 | 40.2 | 39.1 | 42.0 | 41.3 |
|  | EWC Kirkpatrick et al. (2017) | 36.7 | 36.1 | 36.4 | 36.1 | 33.1 | 31.9 | 35.6 | 34.7 |
|  | LwF Li & Hoiem (2017) | 36.4 | 35.8 | 36.7 | 35.7 | 34.3 | 33.5 | 36.1 | 35.4 |
|  | SDC Yu et al. (2020) | 39.2 | 38.8 | 44.5 | 44.1 | 35.2 | 34.1 | 39.4 | 38.1 |
|  | PASS Zhu et al. (2021b) | 39.8 | 39.3 | 45.0 | 44.7 | 37.9 | 35.5 | 39.9 | 38.5 |
|  | IL2A Zhu et al. (2021a) | 43.2 | 42.9 | 50.6 | 50.4 | 43.9 | 39.4 | 44.2 | 43.7 |
|  | TwF Boschini et al. (2022) | 42.7 | 42.1 | 49.4 | 48.9 | 42.3 | 42.1 | 43.6 | 43.3 |
|  | SCoMMER Sarfraz et al. (2023) | 43.8 | 42.5 | 51.1 | 50.6 | 43.4 | 42.3 | 44.9 | 44.1 |
| **FSCIL** | SAVC Song et al. (2023) | 40.0 | 39.6 | 46.8 | 46.3 | 38.3 | 35.9 | 40.1 | 39.6 |
| **LTCIL** | SubProto Wang et al. (2024) | 47.8 | 47.4 | 55.7 | 55.2 | 47.3 | 47.0 | 49.2 | 48.9 |
|  | DEREK (Ours) | **53.1** | **52.5** | **63.3** | **62.1** | **47.4** | **48.0** | **57.9** | **56.7** |
|  |  | 5.3↑ | 5.1↑ | 7.6↑ | 6.9↑ | 0.1↑ | 1.0↑ | 8.7↑ | 7.8↑ |

Table 2: **Imbalance Ratio = 50.** Accuracies by category (LT, CIL, Few-Shot CIL, LTCIL) for 5-task and 10-task setups under ordered and shuffled conditions. Best and second-best scores are highlighted. DEREK consistently outperforms the baselines with improvements denoted by ↑.

$\rho = 10$. These results highlight DEREK's ability to handle varying degrees of class imbalance effectively, demonstrating significant gains, particularly in scenarios with higher imbalance ratios.

**Shuffled LTCIL.** The shuffled LTCIL scenario represents a more challenging and realistic configuration. In this setting, traditional LT and CIL methods show a marked decline in performance compared to the ordered setting. Despite this background, DEREK consistently outperforms the state-of-the-art SubProto method Wang et al. (2024), demonstrating its robustness under stringent conditions. On CIFAR100LT, DEREK demonstrates notable improvements across various imbalance levels. For $\rho = 100$, DEREK achieves an average improvement of +5.15%.

For $\rho = 50$, the improvement is +0.55%, representing a modest gain. For an imbalance ratio of $\rho = 10$, DEREK achieves an improvement of +1.3%. The gains are even more significant on the ImageNet-Subset dataset, where with $\rho = 100$, the improvement reaches +3.75%, while for $\rho = 50$ and $\rho = 10$, DEREK consistently achieves an average improvement of +8.25%. Notably, the performance improvement in the 10-task setting consistently surpasses that in the 5-task setting across various imbalance levels and datasets in both shuffled and ordered configurations. This underscores DEREK's scalability and adaptability. Overall, these results affirm DEREK's effectiveness in handling both ordered and shuffled LTCIL scenarios, consistently setting new benchmarks and outperforming state-of-the-art methods in the most challenging and realistic settings. Furthermore, we do task-wise comparison of DEREK against two variants: baseline, which utilizes same architecture with only cross-entropy loss, and a variant which does only the decoupling part of our method. Finally, we present the performance of DEREK, which integrates both the decoupling and the early knowledge transfer. As illustrated in Figure A4, DEREK exhibits steady accuracy across all tasks, consistently outperforming the baseline by an average margin of 8.7%. This indicates robustness of our approach, effectively maintaining high accuracy even as tasks progress.

**Qualitative Results.** To evaluate the effectiveness of DEREK, we present qualitative results in Figure 3, comparing it to the recent SOTA method, DAKD. We track the predictions of head classes from **task$_0$** across tasks, as they transition to tail classes due to limited memory budget. As shown in Figure 3, DEREK consistently maintains correct predictions for a longer duration. This is attributed to our Early Knowledge

|  | | Ordered | | | | Shuffled | | | |
|---|---|---|---|---|---|---|---|---|---|
|  | | CIFAR | | ImageNet | | CIFAR | | ImageNet | |
|  | Method | 5 | 10 | 5 | 10 | 5 | 10 | 5 | 10 |
| LT | LDAM Cao et al. (2019) | 33.8 | 33.0 | 35.7 | 34.6 | 13.9 | 8.2 | 22.6 | 22.1 |
|  | BalPoE Aimar et al. (2023) | 37.3 | 36.1 | 36.2 | 35.6 | 26.1 | 25.4 | 25.3 | 24.3 |
|  | MDCS Zhao et al. (2023) | 37.0 | 35.4 | 35.9 | 35.5 | 24.3 | 23.7 | 25.6 | 24.1 |
| CIL | iCaRL Rebuffi et al. (2017) | 48.7 | 48.5 | 51.6 | 51.3 | 46.5 | 45.9 | 48.2 | 47.4 |
|  | EWC Kirkpatrick et al. (2017) | 43.0 | 42.1 | 46.7 | 46.1 | 40.6 | 39.7 | 43.8 | 43.6 |
|  | LwF Li & Hoiem (2017) | 43.5 | 42.7 | 47.1 | 46.5 | 41.2 | 41.0 | 44.4 | 43.9 |
|  | SDC Yu et al. (2020) | 45.7 | 45.0 | 48.1 | 47.2 | 42.9 | 42.3 | 45.9 | 45.1 |
|  | PASS Zhu et al. (2021b) | 46.1 | 45.5 | 48.9 | 47.3 | 43.2 | 42.1 | 46.2 | 45.7 |
|  | IL2A Zhu et al. (2021a) | 52.3 | 52.0 | 53.7 | 53.4 | 50.2 | 49.3 | 53.4 | 52.7 |
|  | TwF Boschini et al. (2022) | 51.1 | 51.0 | 52.8 | 52.3 | 49.3 | 48.7 | 52.2 | 51.7 |
|  | SCoMMER Sarfraz et al. (2023) | 52.9 | 52.4 | 54.3 | 53.6 | 49.9 | 49.1 | 52.9 | 52.6 |
| **FSCIL** | SAVC Song et al. (2023) | 46.7 | 46.2 | 50.6 | 49.9 | 43.1 | 42.0 | 48.3 | 47.6 |
| **LTCIL** | SubProto Wang et al. (2024) | 54.6 | 54.3 | 57.4 | 57.2 | 53.6 | 53.1 | 56.2 | 55.4 |
|  | DEREK (Ours) | **57.0** | **57.1** | **68.2** | **67.0** | **54.9** | **54.4** | **64.7** | **63.4** |
|  |  | 2.4↑ | 2.8↑ | 10.8↑ | 9.8↑ | 1.3↑ | 1.3↑ | 8.5↑ | 8.0↑ |

Table 3: **Imbalance Ratio = 10.** Accuracies by category (LT, CIL, Few-Shot CIL, LTCIL) for 5-task and 10-task setups under ordered and shuffled conditions. **Best** and second-best scores are highlighted. DEREK consistently outperforms the baselines with improvements denoted by ↑.

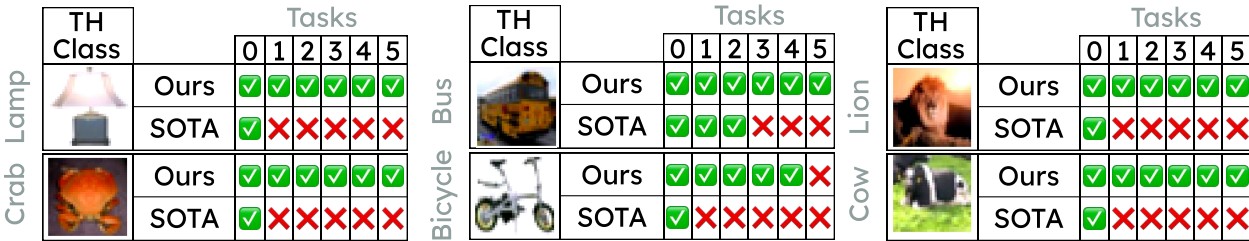

Figure 3: **Prediction consistency of Transitioning Head (TH) classes** on CIFAR100-LT. We track predictions across incremental tasks (0 to 5). A check (✓) indicates a correct prediction, while a cross (✗) denotes misclassification. DEREK maintains robust knowledge of TH classes over time, whereas SOTA suffers from rapid forgetting after the initial task.

Distillation, which preserves the knowledge of representation-rich head classes. Empirical evidence of the performance degradation in Transitioning Heads, averaged across all tasks, is further illustrated in Figure 1 (iii). Additional qualitative results, along with deeper insights into forgetting, are provided in Sec. A.3.

## 5 Ablation Studies

In this section, we evaluate key components of proposed DEREK method. We use CIFAR100-LT with $\rho = 100$ in a shuffled setting for 5 tasks unless stated otherwise.

**Decoupling representations vs. Early Knowledge Distillation.** To analyze the contribution of each component in DEREK, we perform an ablation study and present the results in Table 4. First, we examine the effect of representation decoupling. When using only $L_{head}$, there is no expert specialization, leading to an accuracy drop of $-8.6\%$. In contrast, incorporating $L_{tail}$ to facilitate expertise formation results in a $+4.6\%$ improvement. Next, we assess the impact of performing Early Knowledge Distillation without first establishing dedicated expertise, which, as expected, causes a $-4.5\%$ accuracy drop. Finally, integrating

| Method | $L_{\text{head}}$ | $L_{\text{tail}}$ | $L_{\text{CL}}$ | Accuracy |
|---|---|---|---|---|
| Decoupling | ✓ | ✗ | ✗ | 35.9 (8.6 ↓) |
| Representation | ✓ | ✓ | ✗ | 40.5 (4.0 ↓) |
| Early Knowledge | | | | |
| Distillation | ✗ | ✗ | ✓ | 40.0 (4.5 ↓) |
| DEREK(Ours) | ✓ | ✓ | ✓ | **44.5** |

| Decoupling Representation | Early Knowledge Distillation | Accuracy |
|---|---|---|
| Imbalanced | Imbalanced | 38.8 (5.7 ↓) |
| Balanced | Balanced | 41.7 (2.8 ↓) |
| Balanced | Imbalanced | 35.2 (9.3 ↓) |
| Imbalanced | Balanced | **44.5** (Ours) |

Table 4: Left: Evaluating the impact of different loss components of DEREK on overall accuracy. Right: Analysis of balanced versus imbalanced samplers across different stages of our pipeline. Both are for CIFAR100-LT, shuffled, 5-Task.

all components in DEREK yields the highest accuracy of 44.5%, demonstrating the effectiveness of our approach.

**Data Sampling.** In addition to the loss functions, DEREK leverages different data samplers. As presented in Table 4, we explored various data samplers setups across different parts of our proposed pipeline to verify that the optimal configuration involves using an imbalanced sampler while decoupling, followed by a balanced sampler during Early Knowledge Distillation. This yields a +2.8% accuracy improvement compared to other setups.

**Per-Group Analysis: Quantifying the Role of Transitioning Heads.** To directly isolate the role of TH classes in both accuracy and forgetting, we report per-group metrics (TH vs. Tail) in Table 5. TH forgetting is defined as the average accuracy drop on TH classes across tasks, and Tail forgetting is defined analogously. The TH–Tail gap denotes the difference in forgetting between these two groups.

| | Method | $L_{\text{head}}$ | $L_{\text{tail}}$ | $L_{\text{CL}}$ | Overall Acc ↑ | TH Acc ↑ | Tail Acc ↑ | TH Fgt ↓ | Tail Fgt ↓ | TH–Tail Gap ↓ |
|---|---|---|---|---|---|---|---|---|---|---|
| A | $L_{\text{head}}$ only (Baseline) | ✓ | ✗ | ✗ | 35.9% | 25.7% | 37.3% | 18.7% | 14.3% | +4.5% |
| B | $L_{\text{head}} + L_{\text{tail}}$ (Only Stage 1) | ✓ | ✓ | ✗ | 40.5% | 31.9% | 39.4% | 12.2% | 11.2% | +1.0% |
| C | **DEREK (Stage 1 + Stage 2)** | ✓ | ✓ | ✓ | **44.5%** | **36.1%** | **47.5%** | **8.8%** | 11.8% | **−3.0%** |

Table 5: Per-group accuracy and forgetting across ablation conditions on CIFAR100-LT (Shuffled, 5-Task, $\rho$=100). Acc: accuracy (↑ higher is better); Fgt: forgetting (↓ lower is better).

Starting with accuracy, at baseline (Row A), TH accuracy is 25.7%, compared to 37.3% for Tail classes, indicating that TH classes are already the weaker group before forgetting is considered. Adding Stage 1 (Row B) improves TH accuracy by +6.2%, while Tail classes gain +2.1%, suggesting that the larger relative benefit goes to TH. Under the full method (Row C), TH accuracy reaches 36.1% and Tail accuracy reaches 47.5%.

Turning to forgetting, at baseline (Row A) TH forgetting is 18.7% versus 14.3% for Tail, a gap of +4.5%. This suggests that the degradation at the transition is attributable more to TH classes than to a uniform effect across all classes. We also note that TH classes make up a larger share of the dataset, so the per-class forgetting gap is amplified at the population level.

Under the full DEREK (Row C), TH forgetting falls to 8.8% and the gap moves to −3.0%. We thus observe that the group contributing most to degradation at baseline becomes the better-preserved group once both stages are applied.

For ablations concerning experts, we refer the reader to Sec. 6; for additional ablations on warm-up, freezing, choice of loss for $L_{\text{tail}}$, early vs. late knowledge distillation, specialization vs. ensembling, see Sec. A.2.

# 6 Fairness Analysis

At the heart of DEREK lies a multi-expert architecture, designed to enable Early Knowledge Transfer. Naturally, such a design introduces an increase in parameters. This raises an important concern: *is the comparison fair*? To address this, we break the question into two parts. First, *what is the effect of the expert count on accuracy*? Table 6 makes this immediately clear. With just a single expert ($n = 1$), DEREK already surpasses the state of the art by a healthy margin of $+2.7\%$. This alone establishes a strong lower bound. Increasing the number of experts to $n = 2$ further boosts performance by $+1.6\%$, validating that additional experts reinforce accuracy. Thus, even in its most basic form, DEREK is performant; scaling experts only strengthens the case.

| Method | Train (in Hours) | Parameters (in $10^6$) | Accuracy (%) |
|---|---|---|---|
| DEREK ($n = 1$) | 0.5 | 43.1 | 42.9 |
| DEREK ($n = 2$) | 1.0 (2x) | 86.2 (2x) | 44.5 (1.6 ↑) |
| DAKD | 0.5 (1x) | 21.8 (0.5x) | 40.2 (2.7 ↓) |
| + ↑ epoch | 1.5 (3x) | 21.8 (0.5x) | 40.1 (2.8 ↓) |
| + ↑ param | 1.3 (2.6x) | 60.2 (1.4x) | 23.3 (19.6 ↓) |
| + ↑ epoch, param | 4.2 (8.4x) | 60.2 (1.4x) | 23.5 (19.4 ↓) |

Table 6: Comparison of methods based on training time, inference time, parameters, and accuracy. All multipliers and accuracy deltas are normalized to DEREK ($n = 1$). Experiments on CIFAR100-LT (Shuffled 5-Task, $\rho$=100).

Second, we ask: *how should state-of-the-art methods be scaled to ensure fairness*? One might argue that DEREK has an advantage due to greater training time and parameters. To probe this, we deliberately inflate the competing baseline (DAKD) along two axes: training duration and parameter count. Tripling its training time—equivalent to three times more epochs—yields no gains; in fact, performance slightly declines, a clear signal of overfitting. Enlarging its parameter budget by 1.4× only exacerbates the issue, leading to a sharp drop in accuracy, a signal of underfitting due to optimization instability. Finally, we increase both training time and parameters together in an attempt to counteract these effects, yet the outcome is still worse.

Taken together, these results form consistent narrative. The simplest version of DEREK already outperforms SOTA. Scaling experts amplifies this advantage. Meanwhile, scaling baseline either stagnates or degrades it (refer A.3.1 for more scaling experiments). In other words, DEREK not only starts ahead but also scales gracefully, while alternatives do not. By this measure, comparison is not only fair–it highlights structural strength of DEREK. We also note that the full-data distillation in Stage 2 operates only within task $t$ using the current task's training data. This data is already present in memory for standard training and does not accumulate across tasks. At any point in training, the memory footprint is therefore bounded by the data of a single task, exactly as in standard single-task training. Early Knowledge Distillation introduces no additional memory overhead beyond what is already required to train on the current task.

**Hyperparameters.** A further dimension of fairness lies in how hyperparameters are tuned. A common practice in continual learning is *end-of-training HPO*, where entire data stream is replayed multiple times to select best configuration. While this can boost reported numbers, it is unrealistic–real-world continual learners only see stream once. To avoid this unfair advantage, we follow the more principled *first-task HPO* strategy Lee et al. (2024), where hyperparameters are tuned only on first task and then fixed for all subsequent tasks. This aligns with practical constraints and ensures that no method benefits from hindsight. Under

| HPO Strategy | Accuracy (%) |
|---|---|
| End-of-Training | 45.2 (0.7 ↑) |
| First-Task | 44.5 |

Table 7: Comparison of HPO for DEREK on CIFAR100-LT.

this stricter setting, DEREK consistently converges to 0.5 for all hyperparameters (Table 7), and these fixed values are applied across all reported results. For completeness, we also examined end-of-training HPO, which improved DEREK, but this setting is unfair and not used in our main results.

# 7 Conclusion and Future Work

In this work, we establish *Transitioning Heads Challenge* in LTCIL, where head classes shrink & transition into tail classes in later tasks, suffering accelarating catastrophic forgetting. We proposed DEREK, novel approach that decouples head & tail class learning, enabling *Early Knowledge Distillation*. Extensive experiments validate our hypothesis, showing that DEREK consistently outperforms state-of-the-art methods across multiple benchmarks. A promising future direction is extending LTCIL to multimodal settings. Given ease of text generation compared to image collection, this could reshape our understanding of long-tail distributions and continual learning in multimodal context. It would also be valuable to investigate whether structural degradation in learned representations, such as the loss of class separability and effective dimensionality observed in self-supervised settings Dai et al., also emerges in supervised, discrete-task continual learning settings where TH induces a sudden reduction in available training data.

### Broader Impact Statement

By enabling models to learn new classes incrementally without requiring exhaustive retraining, our work reduces energy consumption, contributing to more sustainable AI practices. Additionally, our work operates under realistic assumptions about data distributions, further enhancing its practicality. Importantly, the improved performance achieved by DEREK compared to existing methods takes a step towards mitigating bias introduced by imbalanced training data. By creating models that are more equitable in their predictions, our work aligns with the goal of fostering responsible AI deployment, positively impacting real-world applications. To the best of our knowledge, our work does not introduce any direct adverse effects. Instead, it contributes to the development of more sustainable and unbiased continual learning systems.

### Acknowledgments

Rahul Vigneswaran would like to thank the Reliance Foundation for their Postgraduate Fellowship. Hari Chandana Kuchibhotla would like to thank MoE for the generous PMRF fellowship support. We thank the anonymous reviewers for their valuable feedback that improved the presentation of this paper.

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
