# A Appendix

Our code will be made publicly available upon acceptance for further research and reproducibility. To preserve the narrative flow of the main paper, we provide additional implementation details and extended experiments in this supplementary material. While not essential for understanding the core content, these additional details and results enhance the overall clarity and depth of our findings.

**Table of Contents**

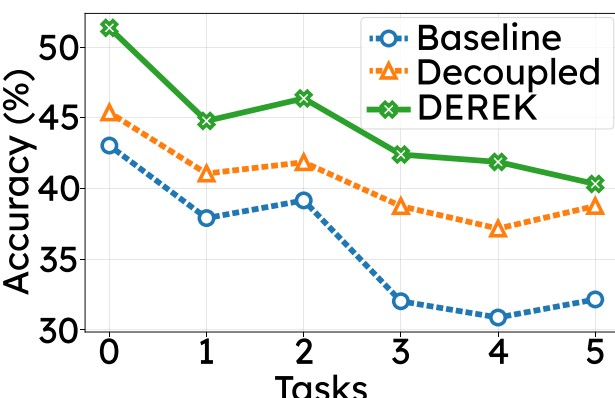

Figure A4: Task-wise accuracy on CIFAR100-LT ($\rho = 100$, shuffled 5-task scenario). DEREK (cross) outperforms the baseline (circle) and decoupled variant (triangle) with an average gain of $+8.7\%$

## A.1  Additional Experimental Details

### A.1.1  Comparison Methods

We benchmark our approach against a comprehensive suite of established CIL methods, including EWC Kirkpatrick et al. (2017), LwF Li & Hoiem (2017), IL2A Zhu et al. (2021a), iCaRL Rebuffi et al. (2017), TwF Boschini et al. (2022), SCoMMER Sarfraz et al. (2023), SDC Yu et al. (2020), PASS Zhu et al. (2021b), IL2M Belouadah & Popescu (2019), BiC Wu et al. (2019), WA Zhao et al. (2020), SSIL Ahn et al. (2021),

MAFDRC Chen & Chang (2023), and FOSTER Wang et al. (2022a). Additionally, we compare the Few-Shot CIL method SAVC Song et al. (2023) and LT methods like LDAM Cao et al. (2019), BalPoE Aimar et al. (2023), and MDCS Zhao et al. (2023). Our evaluation includes comparisons with the latest SOTAs in LTCIL. DAKD He (2024) Single-Staged and EEIL+LWS, FOSTER+LWS, LUCIR+LWS, PODNET+LWS Liu et al. (2022), SubProto Wang et al. (2024) are Two-Staged. Unless otherwise specified, we treat DAKD as the primary state-of-the-art (SOTA) baseline for comparison, as SubProto's code is not publicly available.

### A.1.2   Evaluation Metrics

We adopt the evaluation metrics from Liu et al. (2022); Wang et al. (2024); He (2024). For each task $\textbf{task}_{\textbf{i}}$, accuracy $A_i$ is calculated as:

$$A_i = \frac{\text{Number of correctly classified samples in } \textbf{task}_{\textbf{i}}}{\text{Total number of samples in } \textbf{task}_{\textbf{i}}}$$

To evaluate overall performance, we compute the mean accuracy $\overline{A}$ across all tasks. Given $T$ tasks, $\overline{A}$ is defined as:

$$\overline{A} = \frac{1}{T} \sum_{i=1}^{T} A_i$$

### A.1.3   Complexity Analysis of DEREK

To clarify the distinctions between the decoupling and the early knowledge distillation phases, we employ a switch mechanism in Figure 2 purely for illustrative purposes. This visual aid may give the impression of increased complexity during inference, but it is essential to understand that the switch serves only to differentiate between phases and does not impact the actual flow or complexity of inference.

In Figure 2, the switch is set to position 1 during the decoupling phase and to position 2 during the early knowledge distillation phase. These switches merely control which data loader, losses, and trainable or frozen components are active; they do not alter the inference process itself. Therefore, during inference, these factors are irrelevant, and the process proceeds as follows:

The input data is passed through all the expert heads, $f_j^{\text{head}}$ and $f_k^{\text{tail}}$, for all $j \in n_{head}$ and $k \in n_{tail}$. The outputs from these heads are then processed through $g_j^{\text{head}}$ and $g_k^{\text{tail}}$ and subsequently summed to produce $h^{\text{head}}$ and $h^{\text{tail}}$. These aggregated outputs are then passed through $C^{\text{head}}$ and $C^{\text{tail}}$ to generate the logits. Finally, the logits are averaged with equal weights to produce the final logit, upon which an argmax operation is performed to determine the final prediction.

### A.1.4   Algorithms

The complete warm-up procedure is detailed in Algorithm 2, while the full DEREK procedure is outlined in Algorithm 1.

---

**Algorithm 2** Warm-up

---

**Require:** Tasks: $\mathcal{T}$, Epochs: $e_0$, Dataset: $\mathcal{D}$, Samplers: $\mathcal{S}_{\mathrm{imb}}, \mathcal{S}_{\mathrm{bal}}$, Counts: $n_{\mathrm{head}}, n_{\mathrm{tail}}$, LR: $\eta$, $f$: Feature Extractor, $g$: Linear, $C$: Classifier

---

26 **for** *task* $t \in \mathcal{T}$ **do**
27    **if** $t == 0$ **then**
28      // Warm-up
29      **for** *epoch* $\leftarrow 1$ **to** $e_0$ **do**
30        Sample batch $\mathcal{B} \sim \mathcal{S}_{\mathrm{imb}}(\mathcal{D}_t)$
31        **for** *each* $(x, y) \in \mathcal{B}$ **do**
32          $z \leftarrow C(g(f(x)))$   $\mathcal{L} \leftarrow \mathcal{L}_{\mathrm{CE}}(z, y)$
33          // $\theta_0$: params of $f, g, C$
34          $\theta_0 \leftarrow \theta_0 - \eta \nabla_{\theta_0} \mathcal{L}$
35        **end**
36      **end**
37      // Initialization of Experts
38      $\{f_j^{\mathrm{head}}, g_j^{\mathrm{head}}\}_{j=1}^{n_{\mathrm{head}}} \leftarrow \mathrm{copy}(g, f)$
39      $\{f_k^{\mathrm{tail}}, g_k^{\mathrm{tail}}\}_{k=1}^{n_{\mathrm{tail}}} \leftarrow \mathrm{copy}(g, f)$
40      $C^{\mathrm{head}}, C^{\mathrm{tail}} \leftarrow \mathrm{copy}(C)$
41    **end**
42 **end**

---

## A.2 Additional Ablation Studies

| Modification | Accuracy |
|---|---|
| No Warmup | 37.6 $(6.9 \downarrow)$ |
| No $g$ layer | 43.00 $(1.5 \downarrow)$ |
| No Freezing | 40.8 $(3.7 \downarrow)$ |
| DEREK | **44.5** (Ours) |

Table A8: Removing key components (warmup, trainable $g$ layer, frozen backbone) reduces accuracy. DEREK achieves the highest performance, highlighting their importance. Results on CIFAR100-LT (5-Task).

**Frozen feature space.** In addition to tailored loss functions and a specialized data sampler, we freeze the feature space to prevent forgetting. At a glance, this may seem like a simple step but as demonstrated in Table A8, this is a vital step providing a significant $+3.7\%$ improvement in accuracy.

**Warmup.** As outlined in Algorithm 2, we perform an initial warmup using a single expert, duplicating it to the required number of head and tail experts for decoupling. The results of this ablation, presented in Table A8, show a significant performance increase of $+6.9\%$ over the configuration without warmup, making it a crucial step. Note that, this is only done once at the start of training and never again.

**g layer.** We opt-out of weighting the experts heuristically in favour of an end-to-end trainable non-linear layer $g$. Table A8 indicates that this decision pays off with $+1.5\%$ increase in accuracy over equally weighted experts.

| Method | Overall Acc ↑ | TH Acc ↑ | Tail Acc ↑ |
|---|---|---|---|
| Late KD | 40.3% | 33.3% | 43.6% |
| Early KD (DEREK) | **44.5%** | **36.1%** | **47.5%** |

Table A10: Early vs. Late KD under identical architecture on CIFAR100-LT (Shuffled, 5-Task, $\rho=100$). The two conditions differ only in when distillation is applied.

| $L_{tail}$ | Standalone | DEREK (Ours) |
|---|---|---|
| CB | 14.3 | 44.5 (30.2 ↑) |
| LDAM | 11.5 | 44.8 (33.3 ↑) |

Table A9: Replacing $L_{\text{tail}}$ in DEREK with any long-tailed loss yields similar performance, confirming that TH, not imbalance, is the key challenge. We use Class-Balanced loss and report results on CIFAR100-LT (5-Task).

**Tail Loss ($L_{\text{tail}}$).** While CB loss aids in handling data scarcity for tail classes, it is not the primary driver of DEREK's performance improvements. As demonstrated in Table A9, substituting CB loss with alternative losses such as LDAM Cao et al. (2019) does not significantly impact results. This underscores that DEREK's effectiveness stems from explicitly modeling TH rather than solely mitigating class imbalance. Despite LDAM outperforming CB in some settings, we opt to use CB loss due to its widespread adoption in the literature.

**Early vs. Late Knowledge Distillation.** A natural question is whether the gains of DEREK stem from the *timing* of distillation or simply from having access to more data during the distillation step. In LTCIL, these two factors are difficult to separate because they reflect the same underlying phenomenon: "too late" is precisely defined as "after the data loss event." A data-matched Late KD condition, where distillation at task $t+1$ retains access to all 500 original samples, would violate the fixed-memory constraint that defines the problem. The link between timing and data availability is therefore a structural property of LTCIL, not an artifact of our experimental design.

To provide direct empirical evidence for the timing effect, we compare Early KD (DEREK) with a Late KD variant in which distillation is applied at task $t+1$ using only the available replay exemplars. The architecture and training protocol are identical; only the distillation timing differs. Results are reported in Table A10.

Early KD outperforms Late KD on all three metrics: overall accuracy (+4.2%), TH accuracy (+2.8%), and Tail accuracy (+3.9%). These results are consistent with our hypothesis that the distillation window matters, particularly in LTCIL where timing and data availability are inherently linked.

A related question is why prior methods do not already exploit early distillation. As discussed in Section 3 (paragraph 3, page 4), methods such as DAKD and SubProto cannot apply Early KD because the structural identification problem is unresolved: which classes will transition, who the appropriate teacher is, and when the distillation window opens. Without resolving these questions, there is no principled basis on which to apply early distillation. DEREK's decoupling step creates the teacher–student pair, and the deterministic nature of TH transitions defines the distillation window. The two components are coupled by design, and neither functions without the other.

**Specialization vs. Ensembling.** Since DEREK uses a multi-expert architecture, it is reasonable to ask whether the observed gains come from directed specialization or simply from the capacity benefit of maintaining two experts (i.e., ensembling). We address this by constructing a controlled ensemble baseline and by reporting per-expert accuracy to test for specialization directly.

*Isolating the Effect of Stage 1 Specialization.* Table A11 reports three conditions. Row A is an ensemble baseline: two experts trained with the identical $L_{\text{head}}$ loss under the same imbalanced sampling. The architecture, number of experts, and compute budget are the same as in DEREK; the only difference is that

| | Condition | $L_{\text{head}}$ | $L_{\text{tail}}$ | $L_{\text{CL}}$ | Overall Acc ↑ | TH Acc ↑ | Tail Acc ↑ | TH Fgt ↓ | Tail Fgt ↓ | TH–Tail Gap ↓ |
|---|---|---|---|---|---|---|---|---|---|---|
| A | $L_{\text{head}}$ only (Ensemble baseline) | ✓ | ✗ | ✗ | 35.9% | 25.7% | 37.3% | 18.7% | 14.3% | +4.5% |
| B | $L_{\text{head}} + L_{\text{tail}}$ (Stage 1 only) | ✓ | ✓ | ✗ | 40.5% | 31.9% | 39.4% | 12.2% | 11.2% | +1.0% |
| C | **DEREK (Stage 1 + Stage 2)** | ✓ | ✓ | ✓ | **44.5%** | **36.1%** | **47.5%** | **8.8%** | 11.8% | **−3.0%** |

Table A11: Ensemble baseline (A) vs. Stage 1 specialization (B) vs. full DEREK (C) on CIFAR100-LT (Shuffled, 5-Task, $\rho$=100). Rows A and B share the same architecture and compute; only the loss assignment differs.

| | Head classes | Tail classes |
|---|---|---|
| Head expert | **58.1%** | 10.8% |
| Tail expert | 52.5% | **25.3%** |

Table A12: Per-expert, per-group accuracy at task 0 under condition B (Table A11), before Early KD is applied.

specialization is absent. Row B differs from A in exactly one respect: the loss assignment in Stage 1, where one expert receives $L_{\text{head}}$ and the other receives $L_{\text{tail}}$. Row C adds Early KD on top of Stage 1.

The transition from A to B changes only the loss assignment. The resulting +4.6pp improvement in overall accuracy and the reduction of the TH–Tail gap from +4.5% to +1.0% can therefore be attributed to Stage 1 specialization, with Early KD entirely absent. If the gains came primarily from ensemble capacity, we would expect A and B to perform similarly, yet the +4.6pp gap between them suggests that the loss assignment plays a meaningful role. Moving from B to C introduces Early KD, which further reduces TH forgetting ($12.2\% \rightarrow 8.8\%$) and shifts the gap to $-3.0\%$. Each component thus produces a distinct effect that is not easily explained by ensemble capacity alone.

*Direct Specialization Test: Per-Expert Accuracy.* To test whether the two experts develop group-specific strengths, we report per-expert, per-group accuracy at task 0 under condition B in Table A12. We measure at task 0, before Early KD introduces cross-expert knowledge transfer, so that the results reflect the effect of the loss assignment in isolation.

The results show a cross-over pattern: the head expert performs better on head classes (+5.6%), while the tail expert performs better on tail classes (+14.5%). If the two experts differed only through random initialization diversity, we would expect both to have similar relative orderings across groups, with small random differences in absolute level. The inverted cross-over observed here is more consistent with directed specialization than with initialization variance.

We note that the tail expert still achieves reasonable accuracy on head classes (52.5%). This is expected, since $L_{\text{tail}}$ is a class-balanced loss that increases the weight of tail classes but does not remove head classes from the objective. The tail expert still observes head samples and learns from them, just with reduced emphasis. Specialization in this context does not mean that each expert ignores the other group; it means that each expert becomes relatively stronger on its designated group compared to the other expert.

Taken together, these results suggest that the contribution beyond ensembling lies in the directed loss assignment, which creates structured, task-relevant specialization. Each expert learns to protect a different class group, and this structured specialization, rather than the number of models alone, is what enables the TH-specific gains observed in Table A11.

## A.3 Additional Results

### A.3.1 Effects of Scaling Current SOTA

In addition to the Table 6 from the main paper, we study the impact of DAKD under different backbones.

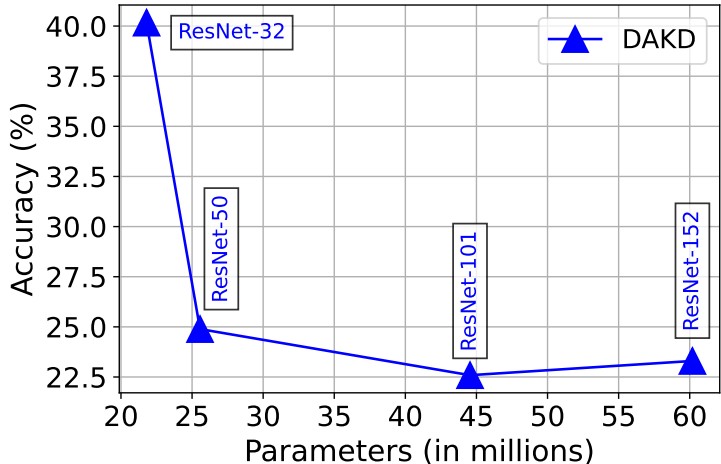

Figure A5: Performance of the current SOTA, DAKD (He (2024)), with progressively larger backbones on CIFAR100LT in a shuffled setting for $\rho = 100$. The plot shows a sharp decline in DAKD's performance as model size increases, consistently remaining below the 44.5% achieved by our proposed method, DEREK. This highlights that increasing the parameter count not only fails to improve accuracy but also has a detrimental effect.

Since the code base for SubProto (2024) is unavailable, we perform this analysis on DAKD (2024). As shown in Figure A5, we evaluate DAKD's performance on progressively larger backbones: ResNet-32 (1x), ResNet-50 (1.2x), ResNet-101 (2x), and ResNet-152 (3x). The results reveal a sharp decline in DAKD's performance with increasing parameters, consistently staying below our method's performance of 44.5%. This degradation can likely be attributed to overfitting, as the downsizing of head classes reduces their representation, while tail classes, despite being fully preserved, have inherently limited data points.

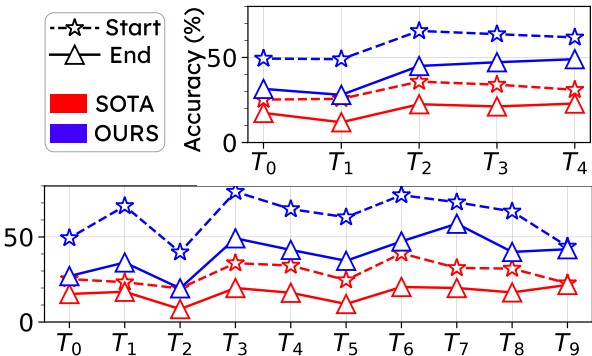

Figure A6: Task-wise forgetting comparison on CIFAR100-LT ($\rho = 100$) under shuffled 5-task and 10-task scenarios. DEREK demonstrates consistently lower forgetting than the state-of-the-art.

### A.3.2 Forgetting Rate Measurement

To evaluate forgetting, we measure the accuracy of each task at two critical points: immediately after the task is learned (*Start*) and after training has been completed on all subsequent tasks (*End*). This allows us to quantify how much performance is retained over time. As illustrated in Figure A6 (for both 5-task and 10-task scenarios with an imbalance factor of 100), DEREK consistently maintains higher *End* accuracy compared to the *Start* accuracy of the state-of-the-art baseline. This significant margin highlights DEREK's robustness to catastrophic forgetting and its effectiveness in long-tailed continual learning settings.

### A.3.3 Learning From Scratch (LFS) setup

Beyond our primary experiments, we conducted a comprehensive analysis of the Learning from Scratch (LFS) paradigm, as introduced by He (2024). Unlike the setup detailed in Section 4.1, where models initially train on 50 classes before progressively incorporating additional classes, the LFS paradigm allocates all classes to tasks from the beginning. This presents a more demanding challenge, requiring models to be trained incrementally without the benefit of an initially learned robust representation from the 50 pre-trained classes.

The results for both the CIFAR100LT and Imagenet-Subset datasets, with an imbalance ratio of $\rho = 100$, are summarized in Table A13 for the 10-task and 20-task shuffled configurations. CIL methods show notably lower performance compared to LTCIL methods. Interestingly, although DEREK lacks specialized mechanisms tailored to this specific challenge, it performs comparably to DAKD (the method that originally introduced this setup). Specifically, in the CIFAR100LT 10-task configuration, DEREK surpasses DAKD by 0.7%. In other configurations, DEREK remains closely competitive with DAKD, with an average performance difference of only 0.8%. This demonstrates DEREK's remarkable effectiveness, even in settings beyond its original scope.

|  | Method | CIFAR | | ImageNet | |
|---|---|---|---|---|---|
|  |  | 10 | 20 | 10 | 20 |
| CIL | IL2M Belouadah & Popescu (2019) | 31.3 | 29.9 | 31.7 | 25.2 |
|  | BiC Wu et al. (2019) | 28.8 | 20.1 | 33.3 | 30.8 |
|  | WA Zhao et al. (2020) | 27.6 | 23.4 | 32.5 | 29.0 |
|  | SSIL Ahn et al. (2021) | 26.0 | 26.1 | 30.3 | 25.9 |
|  | MAFDRC Chen & Chang (2023) | 32.6 | 31.9 | 40.0 | 34.4 |
|  | FOSTER Wang et al. (2022a) | 30.4 | 29.9 | 34.3 | 29.7 |
| LTCIL | EEIL+LWS He (2024) | 33.6 | 32.2 | 36.8 | 30.3 |
|  | FOSTER+LWS He (2024) | 31.2 | 30.6 | 36.4 | 33.9 |
|  | LUCIR+LWS He (2024) | 31.0 | 31.0 | 39.8 | 34.7 |
|  | PODNET+LWS He (2024) | 30.4 | 30.3 | 35.4 | 31.7 |
|  | DAKD He (2024) | 35.6 | **34.5** | **45.1** | **40.7** |
|  | DEREK (Ours) | **36.3** | 33.6 | 44.0 | 40.4 |
|  |  | 0.7↑ | 0.9↓ | 1.1↓ | 0.3↓ |

Table A13: Accuracies across methods by category (CIL, LTCIL, Ours). Evaluations are conducted under LFS shuffled conditions for 10-task and 20-task setups. **Best** and second-best performances are highlighted. Increase and decrease in accuracy relative to the best baseline are marked by ↑ and ↓ respectively.

### A.3.4 Representation Drift

In the decoupling step, our method employs independent experts for head and tail classes to decouple their representations. However, we observe that this decoupling leads to a drift between the experts' feature representations. To study this phenomenon, we conducted an experiment using the CIFAR100LT dataset ($\rho = 100$, shuffled) with a focus on the first 50 classes. In Table A14, $h^{\text{head}}$ and $h^{\text{tail}}$ represent the feature embeddings of the head and tail experts, respectively. To quantify the drift, we calculate the cosine distance between these feature representations, with results highlighted in orange. To mitigate this drift and improve consistency across the experts, we introduce a contrastive loss $L_{\text{Con}}$, formulated as:

$$L_{Con} = \sum_{i=j} \left( 1 - \frac{z_i^{\text{head}} \cdot z_j^{\text{tail}}}{\|z_i^{\text{head}}\|\|z_j^{\text{tail}}\|} \right)$$
$$+ \sum_{i \neq j} \max \left( 0, \frac{z_i^{\text{head}} \cdot z_j^{\text{tail}}}{\|z_i^{\text{head}}\|\|z_j^{\text{tail}}\|} - m \right), \tag{5}$$

where $m$ is the margin that enforces separation between non-matching pairs in the logit space (commonly set to 0.5). As shown in Table A14, the cosine distance between $h_{\text{Con}}^{\text{head}}$ and $h_{\text{Con}}^{\text{tail}}$ (representations after applying $L_{\text{Con}}$) is reduced to 0.0164 (highlighted in blue), compared to 1.891 without $L_{\text{Con}}$, demonstrating a remarkable 115x reduction in drift.

|  | $h_{\text{Con}}^{\text{head}}$ | $h_{\text{Con}}^{\text{tail}}$ | $h^{\text{head}}$ | $h^{\text{tail}}$ |
|---|---|---|---|---|
| $h_{\text{Con}}^{\text{head}}$ | 0.0 | - | - | - |
| $h_{\text{Con}}^{\text{tail}}$ | 0.0164 | 0.0 | - | - |
| $h^{\text{head}}$ | 0.0247 | 0.0398 | 0.0 | - |
| $h^{\text{tail}}$ | 1.8894 | 1.9114 | 1.891 | 0.0 |

Table A14: Cosine distances between feature representations of head and tail experts trained with and without Contrastive loss ($L_{\text{Con}}$) on the CIFAR100LT dataset. The significantly lower distance between $h_{\text{Con}}^{\text{head}}$ and $h_{\text{Con}}^{\text{tail}}$ (highlighted in blue) when compared to the distance between $h^{\text{head}}$ and $h^{\text{tail}}$ (highlighted in orange) demonstrates the effectiveness of $L_{\text{Con}}$ in reducing drift.

### A.3.5 Qualitative Results.

In addition to the qualitative results presented in Sec 5, we provide a more comprehensive analysis in Figure A8. We also examine cases where our method struggles to preserve Transitioning Heads (TH), as shown in Figure A7. Interestingly, classes such as `cloud`, `mouse`, `rabbit`, and `bear` present challenges not only for DEREK but also for other state-of-the-art methods like DAKD. We observed no clear patterns among these failure cases. Notably, `rabbit`, despite being a dominant head class in $\text{task}_0$, is still misclassified. This suggests that the misclassification is primarily due to the inherent difficulty of these classes rather than the Transitioning Heads Challenge itself. However, despite this challenge, our method consistently achieves more correct classifications than DAKD, as shown in Figure A7.

### A.4 Miscellaneous

### A.4.1 Dataset Distribution of typical CIL Datasets

This section outlines the dataset distribution utilized in standard Class Incremental Learning (CIL) experiments. Figures A9a and A9b illustrate the class distributions across various datasets—MNIST, CIFAR10, CIFAR100, and TinyImageNet—divided into 5 and 10 tasks, respectively. Each dataset is meticulously balanced, ensuring equal representation of classes within each task. This uniformity is maintained consistently across all datasets, whether in the 5-task configurations of MNIST and CIFAR10 or the 10-task configuration of TinyImageNet. Although MNIST shows slight imbalance, it remains negligible, closely approximating a balanced distribution. However, this uniform class distribution across tasks may not accurately reflect real-world scenarios where class imbalances are common, potentially limiting the generalizability of CIL methodologies tested on these datasets.

### A.4.2 Sensitivity to Replay Memory Strategies.

The sensitivity of TH to the replay strategy depends on how the memory budget is allocated across classes. We consider three representative cases below.

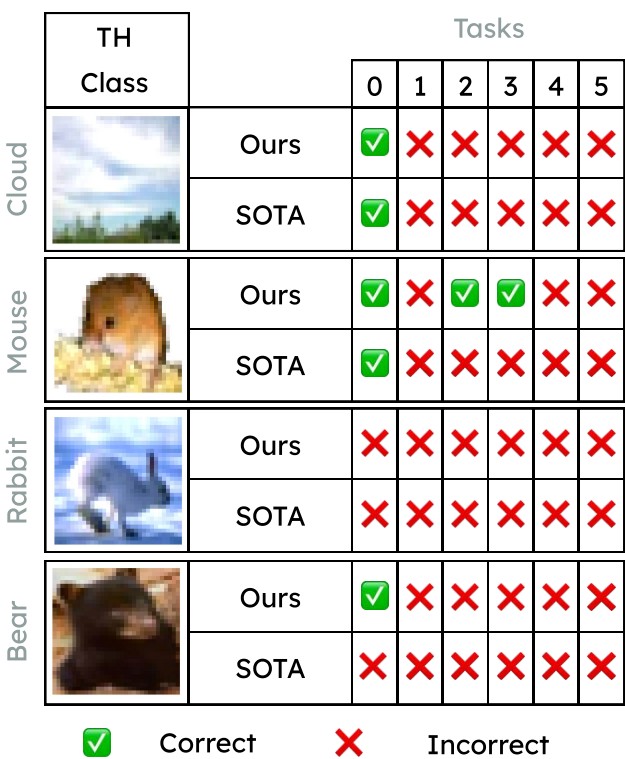

Figure A7: Prediction tracking of a TH class sample across tasks on CIFAR100-LT. Poor performance can be attributed to the inherent difficulty of these classes rather than the Transitioning Heads Conundrum.

**Herding and class-mean exemplar selection (our setting).** These methods determine *which* exemplars are retained, but not *how many* are stored per class. As a result, TH transitions occur at the same rate and magnitude regardless of the selection criterion. The underlying problem is therefore structurally unchanged by this choice.

**Reservoir sampling.** Because head classes dominate the training distribution, reservoir sampling would naturally allocate more buffer slots to them, which can partially reduce the severity of TH transitions. However, tail classes, already underrepresented during training, would receive fewer exemplars and may end up with none in the buffer. In extreme cases, some tail classes could be overwritten entirely. Reservoir sampling thus trades one forgetting problem for another: it reduces TH transitions at the expense of tail representation, which is the trade-off DEREK is designed to avoid.

**Fixed exemplars per class.** This is the only strategy that guarantees tail representation by construction. The drawback is that TH transitions become deterministic and more severe, as head classes move from hundreds of training samples to a fixed exemplar budget in a single step. DEREK addresses this transition through Stage 1 specialization and Early KD, without altering the exemplar budget allocated to tail classes.

We note that our method operates in the fixed-exemplar setting not as a simplifying assumption, but because it is the only replay strategy that preserves tail representation without requiring one class group to subsidize another. As shown in Figure 1(iii), addressing both groups simultaneously without improving one at the expense of the other is the goal that motivates DEREK.

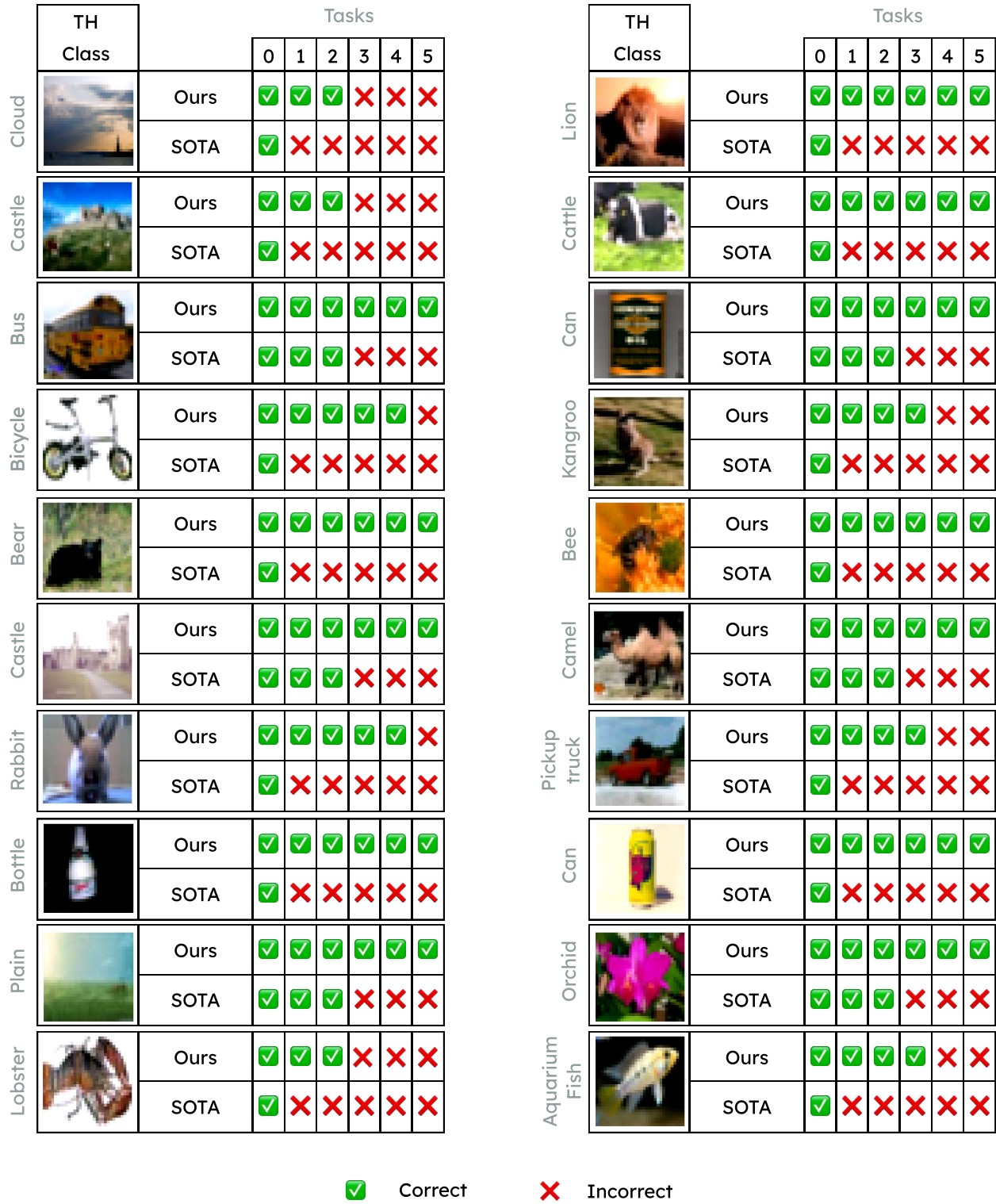

Figure A8: Additional Qualitative Results showing prediction tracking of a TH classes across tasks on CIFAR100-LT. Ours (DEREK) retains correctness longer than DAKD (SOTA).

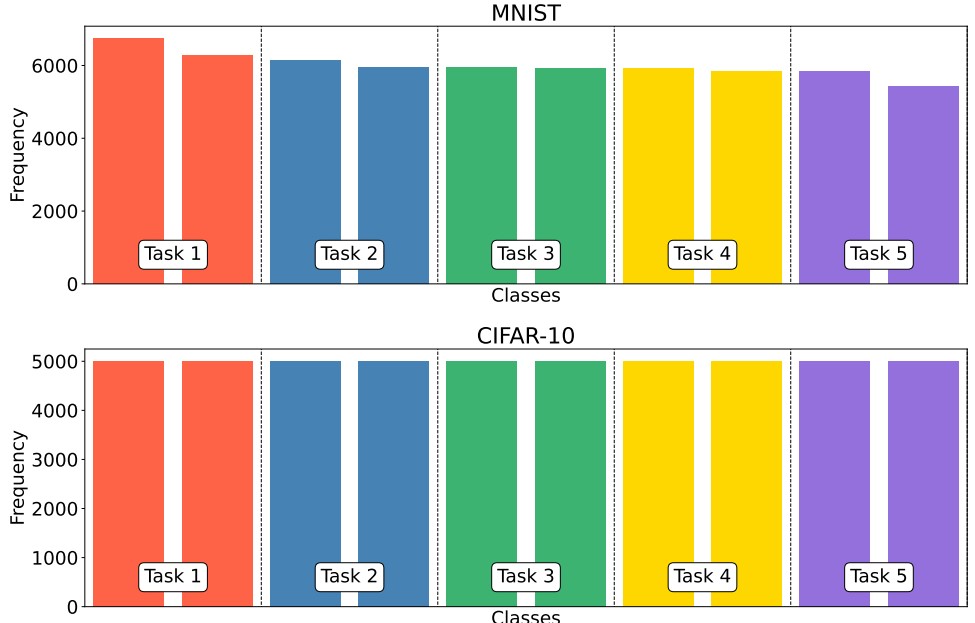

(a) Class distribution of MNIST and CIFAR10 in a 5-task configuration, commonly used in CIL.

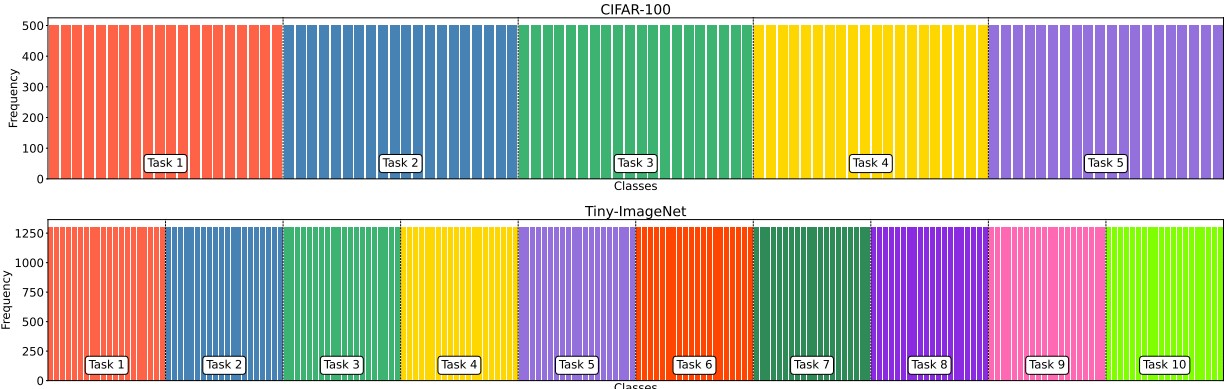

(b) Class distribution of CIFAR100 and TinyImageNet in 5-task and 10-task configurations, respectively.