# OpenReview forum: "Transitioning Heads Conundrum: The Hidden Bottleneck in Long-Tailed Class-Incremental Learning"
_TMLR — Accepted by TMLR_

### Review · Reviewer_uVkU · 2026-01-07

**Summary Of Contributions:**

This paper identifies a previously overlooked phenomenon in Long-Tailed Class-Incremental Learning (LTCIL), termed Transitioning Heads (TH), where initially frequent classes progressively become tail classes due to replay memory constraints, leading to accelerated forgetting. The authors argue that existing methods fail to address this issue because knowledge distillation is applied too late. To mitigate TH forgetting, the paper proposes DEREK, a two-stage approach that decouples head and tail representations and performs Early Knowledge Distillation while full data access is still available. Extensive experiments on CIFAR100-LT and ImageNet-Subset across multiple imbalance ratios and task settings demonstrate consistent improvements over state-of-the-art LTCIL methods.

**Audience:**

Yes

**Audience Explanation:**

The findings of this paper would be of clear interest to at least a subset of the TMLR audience, particularly researchers working on continual learning, class-incremental learning, and learning under long-tailed distributions.

1. The paper identifies Transitioning Heads, a structural and previously underexplored issue that naturally arises in replay-based LTCIL settings. This insight is likely to be relevant to researchers designing or analyzing continual learning systems under realistic memory constraints.

2. The proposed concept of Early Knowledge Distillation, motivated by the temporal availability of data rather than model architecture alone, offers a perspective that may influence how knowledge preservation is approached more broadly in continual and lifelong learning.

**Broader Impact Concerns:**

No.

**Claims And Evidence:**

Yes

**Claims Explanation:**

The main claims of the paper are supported by accurate, convincing, and generally clear empirical evidence.

1. The paper provides clear empirical evidence (e.g., Fig. 1 and task-wise analyses) showing that head classes progressively lose representation under replay constraints and that this transition correlates with accelerated forgetting. This supports the central claim that TH is a structural issue in LTCIL rather than an artifact of a specific method.

2. Extensive results across datasets, imbalance ratios, task orders, and task counts demonstrate consistent and non-trivial improvements. Ablation studies further isolate the contributions of representation decoupling and early distillation, strengthening causal attribution.

3. Comparisons with strong baselines (e.g., DAKD, SubProto) consistently show larger performance degradation on TH classes, supporting the claim that late-stage distillation is insufficient.

**Requested Changes:**

1. Briefly discussing how sensitive TH is to different replay memory allocation strategies will enhance this paper.

2. Provide additional intuition (or empirical comparison with standard KD losses) to better justify its formulation.

3. Add brief intuitive explanations for the two-stage training procedure and the inference-time fusion strategy to improve accessibility for readers less familiar with LTCIL.

---

> ### Author Response · Authors · 2026-03-15
> **Response to Reviewer uVkU (1/3)**
>
> We thank you for your thoughtful feedback. It is encouraging that you find the central claims to be well-supported by the empirical evidence, and that you view TH as a structural issue arising naturally in replay-based LTCIL rather than an artifact of any specific method. We also value your observation that Early KD, by grounding distillation in the temporal availability of data, may offer a perspective relevant beyond the specific setting studied here. We address each of your suggestions below.
>
> ## Q1: Sensitivity to Replay Memory Strategies
>
> > [REQUESTED CHANGE]: "Briefly discussing how sensitive TH is to different replay memory allocation strategies will enhance this paper."
>
> We thank you for this question. The sensitivity of TH to the replay strategy depends on how the memory budget is allocated. We consider three cases.
>
> **Herding and class-mean exemplar selection (our setting):** These methods influence which exemplars are retained, but not how many are stored per class. As a result, TH transitions occur at exactly the same rate and magnitude. The underlying problem is therefore structurally unchanged by the choice of selection criterion.
>
> **Reservoir sampling:** Because head classes dominate the training distribution, reservoir sampling would naturally allocate more buffer slots to them. This can partially reduce the severity of TH transitions. However, this comes at a cost. Tail classes, which are already underrepresented during training, would receive fewer exemplars and may even end up with none in the buffer. In extreme cases, some tail classes could be overwritten entirely. Reservoir sampling therefore replaces one forgetting problem with another. It reduces TH at the expense of tail representation, which is precisely the trade-off DEREK is designed to avoid.
>
> **Fixed exemplars per class:** This strategy guarantees tail representation by design. The drawback is that TH transitions become deterministic and severe. Head classes move from hundreds of training samples to a fixed exemplar budget in a single step. DEREK addresses this transition directly through Stage 1 specialization and Early KD, without altering the exemplar budget allocated to tail classes.
>
> Our method is therefore designed for the fixed-exemplar setting not as a simplifying assumption, but because it is a replay strategy that preserves tail representation without compromise. As shown in Figure 1(iii), addressing both class groups simultaneously, without improving one at the expense of the other, is exactly the goal of DEREK. We have added this discussion to Section 4.1 in the revised paper.
>
> We have added a section discussing the same in the revised paper in Appendix A.4.2 (pages 8-9).
>
> ---

---

> ### Author Response · Authors · 2026-03-15
> **(cont.) Response to Reviewer uVkU (2/3)**
>
> ## Q2: Intuition for the LCL Loss Formulation
>
> > [REQUESTED CHANGE]: "Provide additional intuition (or empirical comparison with standard KD losses) to better justify its formulation."
>
> We thank you for raising this point. We had described in Section 3.2 (page 6) of the submitted paper an emergent CE property of LCL. What we had not explicitly stated, and clarify here, is why this property matters specifically in the LTCIL setting (which is the motivation) and why KL-based KD is not an adequate substitute. We have added this motivation to Section 3.2 (page 6) in the revised paper.
>
> The emergent CE property means that as the two experts align, LCL naturally converges to a CE form. This behavior is intentional rather than incidental. Prior work in long-tail learning has shown that fine-tuning classifiers with balanced sampling effectively corrects the bias introduced by imbalanced training (Kang et al., 2020). In Stage 2, DEREK uses balanced sampling, which introduces competing gradients across all class groups at the same time. KL becomes inactive once the two models align, so it provides no gradient to maintain that alignment when other objectives are present. In contrast, the convergence of LCL to CE continues to reinforce the tail classifier during the balanced phase even after distillation has completed. This allows knowledge transfer and classifier correction to occur within the same objective.
>
> **Empirical comparison.** We compared LCL with KL as the inter-expert distillation loss while keeping all other components unchanged.
>
> | Loss | Overall Acc ↑ | TH Acc ↑ | Tail Acc ↑ |
> |---|---|---|---|
> | LCL (Ours) | 44.5% | 36.1% | 47.5% |
> | KL | 38.0% | 32.6% | 37.8% |
>
> **Table Q2-A:** LCL vs. KL as the inter-expert distillation loss under identical architecture and training protocol, differing only in the distillation objective.
>
> As shown in Table Q2-A, replacing LCL with KL reduces TH accuracy by 3.5% but reduces Tail accuracy by 9.7%, which is almost three times larger. TH classes are the direct target of the distillation signal, so both losses transfer knowledge to them and the difference is modest. Tail classes receive no direct distillation signal. Their performance depends on how stable the classifier remains after alignment. KL does not provide a gradient to maintain this alignment, while LCL does so through its emergent CE behavior.
>
> **Proposed addition to Section 3.2** *(included after “maintaining stability and consistency in knowledge transfer”):*
>
> > This convergence is particularly well suited to the long-tail setting. Prior work has shown that fine-tuning classifiers with balanced sampling can correct the representational bias introduced by imbalanced training (Kang et al., 2020). The emergent CE property of LCL ensures that once knowledge transfer from the head expert is complete, the tail classifier naturally transitions into this balanced fine-tuning regime. As a result, distillation and classifier correction are combined within a single objective.

---

> ### Author Response · Authors · 2026-03-15
> **(cont. to above) Response to Reviewer uVkU (3/3)**
>
> ## Q3: Accessibility of Two-Stage Procedure and Inference Fusion
>
> > [REQUESTED CHANGE]: "Add brief intuitive explanations for the two-stage training procedure and the inference-time fusion strategy to improve accessibility for readers less familiar with LTCIL."
>
> We thank you for this suggestion. Following your recommendation, we prepared a concrete walkthrough of the two-stage procedure, which we have added to Section 3 in the revised paper to aid readers who may be less familiar with the LTCIL setting. We include it here for reference.
>
> **Addition to the end of Section 3.1 (page 6):**
>
> > To illustrate, consider the class *Lamp*, which is a head class in task *t* with 500 samples under ρ=100. During Stage 1, both experts are trained simultaneously on task *t*'s data. The head expert is trained with L_head (standard CE) and learns a rich representation of *Lamp* from all 500 samples, biased naturally toward high-frequency classes. The tail expert is trained with L_tail (CB loss) and learns representations better suited to lower-frequency categories. By the end of Stage 1, the head expert holds a strong, data-rich representation of *Lamp* while the tail expert is better calibrated for the tail regime that *Lamp* is about to enter.
>
> **Addition to the end of Section 3.2 (page 6):**
>
> > Continuing the *Lamp* example: at the end of task *t*, we know deterministically that *Lamp* will become a tail class in task *t+1*, where only a small number of exemplars will be retained. The head expert, having seen all 500 samples of *Lamp*, is designated as the teacher and the tail expert as the student. L_CL transfers this knowledge before the data disappears. When task *t+1* begins and *Lamp* is now managed by the tail expert, the transferred knowledge allows it to handle *Lamp* far more effectively than if it had encountered it for the first time with only a handful of exemplars.

---

### Review · Reviewer_VM9W · 2026-01-13

**Summary Of Contributions:**

This paper studies Long-Tailed Class-Incremental Learning (LTCIL) and identifies Transitioning Heads (TH), i.e., classes that are head classes in early tasks but effectively become tail-like later due to limited replay memory. The authors argue that existing methods apply knowledge distillation too late to prevent this degradation. They propose DEREK, which decouples head/tail representations and performs early distillation to preserve information before head classes transition. Experiments on CIFAR100-LT and ImageNet-Subset across several LTCIL settings show improvements over many baselines.

**Audience:**

Yes

**Audience Explanation:**

LTCIL is a relevant setting, and the TH perspective on replay-induced degradation is potentially useful. The proposed early-distillation/decoupling approach is intuitive and empirically strong across common benchmarks and settings, making it likely interesting to the continual learning and long-tail communities.

**Claims And Evidence:**

No

**Claims Explanation:**

While the empirical results are broad and generally convincing at the aggregate level, key claims are not fully isolated or substantiated:
1. TH is shown as a useful lens, but the paper does not quantify how much overall degradation (or improvement) is specifically attributable to TH

2. The claim that KD “happens too late” is not tested by a controlled early-vs-late distillation comparison under matched architecture/protocol, and may be confounded with differences in data availability

3. The analysis that the two branches learn distinct representations rather than benefiting mainly from ensembling is not well supported

4. The proposed distillation loss is insufficiently motivated and not directly compared to standard KL-based KD.

**Requested Changes:**

Major:

1. Quantify the role of TH: report per-group results (TH vs non-TH) and analyses that better disentangle whether TH drives the observed degradation/improvements.

2. Isolate distillation timing: add controlled early vs late KD comparisons under the same architecture/protocol, controlling for effective data size/usage.

3. Validate decoupling: include representation similarity/diversity analyses (e.g., CKA/cosine) and clarify the contribution beyond ensembling.

4. Justify loss design: motivate the proposed distillation loss and compare directly with standard KL-based KD baselines.

Minor

5. Fix typos (e.g., “maximum fequency” → “maximum frequency” on Page 6).

6. A recent paper [1] studies the performance degradation from the representation structure perspective; it would be interesting to provide more discussion from the representation perspective.

[1] Exploring Structural Degradation in Dense Representations for Self-supervised Learning, NeurIPS 2025.

---

> ### Author Response · Authors · 2026-03-15
> **Response to Reviewer VM9W (1/6)**
>
> We thank you for your thoughtful and detailed feedback. We appreciate your recognition that LTCIL is a relevant problem setting, that the Transitioning Heads perspective offers a useful lens on replay-induced degradation, and that the proposed approach is both intuitive and empirically strong. Each of your requested changes has helped us strengthen the paper with more precise evidence, and we address them individually below.
>
> ## Q1: Quantify TH Role
> > [WEAKNESS]: "TH is shown as a useful lens, but the paper does not quantify how much overall degradation (or improvement) is specifically attributable to TH"
>
> > [REQUESTED CHANGE]: "Quantify the role of TH: report per-group results (TH vs non-TH) and analyses that better disentangle whether TH drives the observed degradation/improvements."
>
> Thank you for this precise request, as the additional findings from this analysis help strengthen our claim by directly isolating the role of TH classes in both accuracy and forgetting.
>
> In this response, we:
> - **Report per-group metrics** (TH vs Tail accuracy and forgetting) across the main ablation conditions (Table Q1-A).
> - **Identify which class group drives degradation at baseline** using the TH–Tail forgetting gap.
> - **Test whether DEREK’s improvements are TH-targeted**.
>
>
> |  | Condition | L_head | L_tail | L_CL | Overall Acc ↑ | TH Acc ↑ | Tail Acc ↑ | TH forgetting ↓ | Tail forgetting ↓ | TH−Tail gap ↓ |
> |---|---|---|---|---|---|---|---|---|---|---|
> | A | L_head only (Baseline) | ✓ | ✗ | ✗ | 35.9% | 25.7% | 37.3% | 18.7% | 14.3% | +4.4% |
> | B | L_head + L_tail (Only Stage 1) | ✓ | ✓ | ✗ | 40.5% | 31.9% | 39.4% | 12.2% | 11.2% | +1.0% |
> | C | Early KD only (Only Stage 2) | ✗ | ✗ | ✓ | 40.0% | 31.6% | 43.8% | 12.2% | 6.9% | +5.3% |
> | D | **DEREK (Stage 1 + Stage 2)** | ✓ | ✓ | ✓ | **44.5%** | **36.1%** | **47.5%** | **8.8%** | 11.8% | **−3.0%** |
>
> **Table Q1-A:** Forgetting delta per ablation condition (↓ lower is better). This table corresponds to Table 4 in page 10, but now includes per-group results and additional metrics.
>
> We report the per-group metrics in Table Q1-A. We begin with the accuracy perspective in these results. At baseline (Row A), TH accuracy (25.7%) is substantially lower than Tail accuracy (37.3%). This gap indicates that TH classes are already underperforming before considering forgetting. Introducing Stage 1 alone (Row B) increases TH accuracy by +6.2%, compared to a +2.1% gain for Tail classes. With the full method (DEREK), TH accuracy improves further to 36.1%, while Tail accuracy reaches 47.5%. These results show that the largest relative improvement occurs for TH classes, which are the group initially performing worst.
>
> We now examine forgetting in Table Q1-A. At baseline (Row A), TH forgetting (18.7%) is substantially higher than Tail forgetting (14.3%), a gap of +4.4%. This shows that TH classes have a key effect on degradation at the transition, rather than a general continual learning effect affecting all classes equally. The issue is further amplified by class composition, since TH classes constitute a much larger portion of the dataset than Tail classes. As a result, the TH forgetting problem is doubly penalized: these classes forget more and they represent the majority of classes.
>
> With the full DEREK method (Row D), the TH−Tail gap becomes −3.0%. TH classes (8.8% forgetting) now forget substantially less than Tail classes (11.8%). The gap does not simply close; it reverses. The class group that was responsible for most of the degradation at baseline becomes the better preserved group under the full method.
>
> We have included this table and the corresponding analysis in the revised paper in Section 5, page 11.

---

> ### Author Response · Authors · 2026-03-15
> **(cont.) Response to Reviewer VM9W (2/6)**
>
> ### But is this improvement TH-specific by design?
>
> The results above corroborate our view that TH drives the degradation and that DEREK addresses it. This raises a natural follow-up question: is DEREK’s improvement selectively targeted at TH classes, or does the method simply improve all classes? Table Q1-A provides a direct way to examine this. Rows C and D both use Early KD; the only difference is whether Stage 1 is applied before it.
>
> **Early KD without specialization (Row C, Table Q1-A):** TH forgetting drops from 18.7% to 12.2% (−6.5%). However, Tail forgetting drops even more, from 14.3% to 6.9% (−7.4%). As a result, the TH−Tail gap actually worsens, increasing from +4.4% to +5.3%. Without Stage 1, both experts are trained in the same way, so Early KD acts as generic distillation without a TH-specific signal. In this setting, it benefits Tail classes more than TH classes.
>
> **Early KD with specialization, i.e., DEREK (Row D, Table Q1-A):** TH forgetting drops from 18.7% to 8.8% (−9.9%), while Tail forgetting decreases more modestly, from 14.3% to 11.8% (−2.5%). The TH−Tail gap flips from +4.4% to −3.0%. Once Stage 1 creates a specialized teacher–student pair, the forgetting reduction produced by Early KD becomes strongly TH-targeted.
>
> The same distillation mechanism either worsens the TH−Tail gap (C: +4.4% → +5.3%) or reverses it (D: +4.4% → −3.0%), depending solely on whether Stage 1 precedes it. Stage 1 decoupling builds expert specialization, and Stage 2 distillation transfers that specialization. Without Stage 1, the same Early KD operation works against the TH claim.
>
>
> ---
>
> ## Q2: Distillation Timing
> > [WEAKNESS]: "The claim that KD “happens too late” is not tested by a controlled early-vs-late distillation comparison under matched architecture/protocol, and may be confounded with differences in data availability"
>
> > [REQUESTED CHANGE]: "Isolate distillation timing: add controlled early vs late KD comparisons under the same architecture/protocol, controlling for effective data size/usage."
>
> Thank you for this point. In this response, we:
> - **Clarify why timing and data availability may be difficult to separate in LTCIL**.
> - **Provide a controlled empirical comparison** between Early KD and Late KD under the same architecture and training protocol (Table Q2-A).
> - **Explain why prior methods cannot apply early distillation**.
>
> In LTCIL, timing and data availability are not easily separable, and describe the same underlying phenomenon. In this setting, “too late” is precisely defined as “after the data loss event”. A data-matched Late KD condition, where distillation is performed at task t+1 with access to the full 500 samples, is not a valid experimental setup under the LTCIL protocol. It would require retaining the entire training data indefinitely, which violates the fixed-memory constraint that defines the problem.
>
> This confound between timing and data availability is thus not an artifact of our experimental design, but a structural property of LTCIL that motivates this work. Early KD is useful precisely because it operates during the phase where both full training data and the need for knowledge transfer coexist.
>
> That said, we provide direct empirical evidence for the timing effect. We compare Early KD (DEREK) with a Late KD condition in which distillation is applied at task t+1 using only the available replay exemplars. The architecture is identical and only the timing differs. The results are reported in Table Q2-A.
>
> | Method | Overall Acc ↑ | TH Acc ↑ | Tail Acc ↑ |
> |---|---|---|---|
> | Early KD (DEREK) | 44.5% | 36.1% | 47.5% |
> | Late KD | 40.3% | 33.3% | 43.6% |
>
> **Table Q2-A:** Early vs. Late KD under identical architecture, differing only in distillation timing.
>
>
> As shown in Table Q2-A, Early KD outperforms Late KD on all three metrics: overall accuracy (+4.2%), TH accuracy (+2.8%), and Tail accuracy (+3.9%). These results are consistent with our hypothesis that the distillation window matters, particularly in LTCIL where timing and data availability are inherently linked.

---

> ### Author Response · Authors · 2026-03-15
> **(cont. to above) Response to Reviewer VM9W (3/6)**
>
> If early distillation were simply a matter of using available data, a natural question is why prior methods do not already exploit it. As noted in Section 2 (paragraph 4, page 3) of the revised paper: *"distillation in these methods remains limited, as they rely on small set of exemplars and fail to address TH. Moreover, their tightly coupled head and tail representations hinder early knowledge distillation."* To make this reasoning more explicit, the core difficulty is that three structural questions need to be resolved before early distillation can be applied: which classes will transition, who should serve as the teacher, and when the distillation window opens. Methods such as DAKD and SubProto do not have a mechanism to resolve these within their frameworks, which is what prevents them from exploiting the early distillation window. DEREK addresses this by using decoupling to create the teacher–student pair and leveraging the deterministic transition to define the distillation window. The two components complement each other by design, and together they make early distillation both well-defined and practically applicable. We have included this elaboration in Section 3 (paragraph 3, page 5) in the revised paper for clarity.
>
> We also note that R-uVkU specifically highlighted "Early KD, motivated by the temporal availability of data rather than model architecture alone" as a perspective that may broadly influence knowledge preservation approaches, which we believe captures our contribution.
>
> We have included this table and the corresponding analysis in the revised paper in Appendix A.2, page 4.
>
> ---
>
> ## Q3: Specialization vs Ensembling
> > [WEAKNESS]: "The analysis that the two branches learn distinct representations rather than benefiting mainly from ensembling is not well supported"
>
> > [REQUESTED CHANGE]: "Validate decoupling: include representation similarity/diversity analyses (e.g., CKA/cosine) and clarify the contribution beyond ensembling."
>
> Thank you again, each of these questions allows us to clarify our contributions more deeply. In this response, we:
> - **Use the ensemble baseline as a controlled comparison**, where two experts exist but specialization is absent.
> - **Show the isolated effect of Stage-1 specialization**.
> - **Provide a direct specialization test** via per-expert, per-group accuracy before EKD.
>
> In Table Q3-A, the ensemble control corresponds to row A. In this setting, both experts are trained with the identical L_head loss using the same imbalanced sampling. The architecture, two-expert structure, and compute budget are identical. What is absent is specialization. Row B differs from A in exactly one respect: the loss assignment in Stage 1. One expert receives L_head while the other receives L_tail. Specialization is therefore the only variable introduced in row B.
>
> | | Condition | L_head | L_tail | L_CL | Overall Acc ↑ | TH Acc ↑ | Tail Acc ↑ | TH forg. ↓ | Tail forg. ↓ | TH−Tail gap ↓ |
> |---|---|:---:|:---:|:---:|---|---|---|---|---|---|
> | A | L_head only (ensemble baseline) | ✓ | ✗ | ✗ | 35.9% | 25.7% | 37.3% | 18.7% | 14.3% | +4.4% |
> | B | L_head + L_tail (Stage 1 only) | ✓ | ✓ | ✗ | 40.5% | 31.9% | 39.4% | 12.2% | 11.2% | +1.0% |
> | D | **DEREK full** | ✓ | ✓ | ✓ | 44.5% | 36.1% | 47.5% | 8.8% | 11.8% | −3.0% |
>
> **Table Q3-A:** Ensemble baseline (A) vs. Stage 1 specialization (B) vs. full DEREK (D). Rows A and B differ only in the loss assignment, isolating the effect of specialization.
>
> The transition from A to B changes only the loss assignment. The resulting +4.6% improvement in overall accuracy and the reduction of the TH−Tail gap from +4.4% to +1.0% therefore reflect Stage 1 specialization alone, with EKD entirely absent. If the gains came purely from ensemble capacity, A and B would perform similarly, yet the +4.6pp gap between them suggests otherwise. Moving from B to D introduces EKD, which further reduces TH forgetting (12.2% → 8.8%) and flips the gap to −3.0%. Each component produces a distinct and interpretable effect, which is not easily explained by ensemble capacity alone.

---

> ### Author Response · Authors · 2026-03-15
> **(cont. to above) Response to Reviewer VM9W (4/6)**
>
> Regarding the suggestion on studying whether the two branches learn distinct representations, we report per-expert, per-group accuracy at Stage 1 in Table Q3-B, which directly studies this issue.
>
> We measure at task 0, before EKD introduces cross-expert knowledge transfer, to isolate the effect of the loss assignment. After EKD is applied, the experts are no longer independent, so task 0 provides the cleanest view of what Stage 1 alone produces.
>
> If specialization occurs, each expert should perform better on its designated class group. If the gains came only from ensemble diversity, including random initialization diversity present in both A and B, both experts would show broadly similar per-group behavior with small random differences.
>
> | | Head classes | Tail classes |
> |---|---|---|
> | Head expert | 58.1% | 10.8% |
> | Tail expert | 52.5% | 25.3% |
>
> **Table Q3-B:** Per-expert, per-group accuracy at task 0 under condition B, before EKD is applied.
>
> The per-expert accuracy in Table Q3-B exhibits a cross-over pattern. The head expert performs better on head classes (+5.6%), while the tail expert performs better on tail classes (+14.5%). Each expert is therefore stronger on its designated group. This asymmetry is difficult to attribute to random initialization diversity, which would produce two experts with similar per-group profiles rather than systematically inverted ones. The observed cross-over is more consistent with directed specialization.
>
> One might ask why the tail expert still achieves reasonable accuracy on head classes (52.5%). This is expected, as L_tail is a class-balanced loss. It increases the weight of tail classes but does not remove head classes from the data or the objective. The tail expert still observes head samples and learns from them, just with less emphasis. Specialization here does not mean that each expert ignores the other group. It means that each expert becomes relatively stronger on its designated group compared to the other expert. Random initialization diversity would produce experts with the same relative ordering across groups at slightly different absolute levels, rather than the inverted cross-over observed here.
>
> The key contribution beyond ensembling is therefore the directed loss assignment, which creates structured and task-relevant specialization. Each expert learns to protect a different class group. This structured specialization, rather than the number of models alone, drives the TH-specific gains.
>
> We have included these tables and the corresponding analysis in the revised paper in Appendix A.2, pages 4-5.
>
> ---

---

> ### Author Response · Authors · 2026-03-15
> **(cont. to above) Response to Reviewer VM9W (5/6)**
>
> ## Q4: Justifying the LCL Loss
> > [WEAKNESS]: "The proposed distillation loss is insufficiently motivated and not directly compared to standard KL-based KD."
>
> > [REQUESTED CHANGE]: "Justify loss design: motivate the proposed distillation loss and compare directly with standard KL-based KD baselines."
>
> We thank you for raising this point. We described in Section 3.2 (page 6) of the submitted paper an emergent CE property of LCL. What we had not explicitly stated, and clarify here, is why this property matters specifically in the LTCIL setting (which is the motivation) and why KL-based KD is not an adequate substitute. We have added this motivation to Section 3.2 (page 6) in the revised paper.
>
> The emergent CE property means that as the two experts align, LCL naturally converges to a CE form. This behavior is intentional rather than incidental. Prior work in long-tail learning has shown that fine-tuning classifiers with balanced sampling effectively corrects the bias introduced by imbalanced training (Kang et al., 2020). In Stage 2, DEREK uses balanced sampling, which introduces competing gradients across all class groups at the same time. KL becomes inactive once the two models align, so it provides no gradient to maintain that alignment when other objectives are present. In contrast, the convergence of LCL to CE continues to reinforce the tail classifier during the balanced phase even after distillation has completed. This allows knowledge transfer and classifier correction to occur within the same objective.
>
> **Empirical comparison.** We compared LCL with KL as the inter-expert distillation loss while keeping all other components unchanged.
>
> | Loss | Overall Acc ↑ | TH Acc ↑ | Tail Acc ↑ |
> |---|---|---|---|
> | LCL (Ours) | 44.5% | 36.1% | 47.5% |
> | KL | 38.0% | 32.6% | 37.8% |
>
> **Table Q4-A:** LCL vs. KL as the inter-expert distillation loss under identical architecture and training protocol, differing only in the distillation objective.
>
> As shown in Table Q4-A, replacing LCL with KL reduces TH accuracy by 3.5% but reduces Tail accuracy by 9.7%, which is almost three times larger. TH classes are the direct target of the distillation signal, so both losses transfer knowledge to them and the difference is modest. Tail classes receive no direct distillation signal. Their performance depends on how stable the classifier remains after alignment. KL does not provide a gradient to maintain this alignment, while LCL does so through its emergent CE behavior.
>
> **Proposed addition to Section 3.2** *(included after “maintaining stability and consistency in knowledge transfer”):*
>
> > This convergence is particularly well suited to the long-tail setting. Prior work has shown that fine-tuning classifiers with balanced sampling can correct the representational bias introduced by imbalanced training (Kang et al., 2020). The emergent CE property of LCL ensures that once knowledge transfer from the head expert is complete, the tail classifier naturally transitions into this balanced fine-tuning regime. As a result, distillation and classifier correction are combined within a single objective.

---

> ### Author Response · Authors · 2026-03-15
> **(cont. to above) Response to Reviewer VM9W (6/6)**
>
> ## Q5: Minor changes
> > Fix typos (e.g., “maximum fequency” → “maximum frequency” on Page 6).
>
> > A recent paper [1] studies the performance degradation from the representation structure perspective; it would be interesting to provide more discussion from the representation perspective.
>
> **On the typo:** We thank you for catching this. The typo on Section 4.1 in Page 7 ("maximum fequency" → "maximum frequency") has been corrected in the revision.
>
> **On Dai et al. (NeurIPS 2025):** This is an interesting pointer, and we thank the reviewer for bringing it to our attention. This is a concurrent work that appeared during our submission process, and the representational perspective it proposes is compelling. Their argument that class separability and effective dimensionality can capture degradation that surface-level metrics miss raises an interesting question about how TH-induced data loss might affect the representation structure of transitioning classes.
>
> That said, the motivation behind DSE does not transfer directly to our setting. DSE is a label-free structural estimator, which is necessary in SSL because ground-truth labels are unavailable. In our supervised setting, we have direct access to labels and can therefore measure the quantities of interest directly: per-class accuracy, per-group forgetting, and the TH−Tail gap. These metrics capture exactly the behavior we want to analyze and are already reported throughout the paper. Using a structural proxy in this context would move us one step away from the signal of interest rather than closer to it.
>
> There is also an important difference in the underlying phenomena. SDD arises from extending self-supervised training over time, whereas TH is caused by a sudden reduction in available training data. Investigating whether similar structural degradation emerges in supervised, discrete-task continual learning settings is an interesting direction, and we have added this to Section 7 (Conclusion and Future Work, page 13) in the revised paper.

---

### Review · Reviewer_FFXq · 2026-02-20

**Summary Of Contributions:**

Summary:

The paper identifies the head-to-tail transition as a critical challenge in long-tailed class-incremental learning and proposes an early knowledge distillation framework to mitigate severe catastrophic forgetting caused by this transition.


Strengths:

S1. The proposed early knowledge distillation framework is simple yet effective.

S2. Experimental results demonstrate that the proposed framework consistently outperforms state-of-the-art methods in long-tailed class-incremental learning settings.


Weaknesses:

W1. While the paper frames the head-to-tail transition as a challenge specific to long-tailed class-incremental learning, similar performance degradation due to the shift from full-data access to limited replay buffer has long been observed in standard class-incremental learning settings.

W2. The proposed framework relies on the assumption that classes in the current task are data-rich while classes from previous tasks become data-scarce due to fixed-memory replay, leading to an inevitable head-to-tail transition. While this assumption is valid under standard class-incremental benchmarks with standard data sampling strategy for replay buffers, its applicability may be limited in real-world data distributions.

W3. While early knowledge distillation is effective, the core idea of transferring knowledge prior to data scarcity does not appear particularly novel, as it follows naturally from existing knowledge distillation paradigms. In addition, the proposed framework incurs increased computational and memory costs due to full-data knowledge distillation and multi-expert modeling.

**Audience:**

Yes

**Audience Explanation:**

The paper focuses on long-tailed class-incremental learning, which exhibits more severe imbalance than standard class-incremental learning and can therefore be considered a more realistic setting. However, the proposed technique itself is not particularly novel and appears to rely on the assumption that head-to-tail transitions inevitably occur, which may limit its applicability.

**Broader Impact Concerns:**

The claim that the method operates under realistic data distribution assumptions may be overstated, and the paper addresses fairness only in terms of class-level performance imbalance rather than broader societal or demographic biases.

**Claims And Evidence:**

Yes

**Claims Explanation:**

The paper clearly demonstrates that the number of transitioning head classes increases over time and these classes suffer from severe catastrophic forgetting. In addition, experimental results show that the proposed framework consistently achieves strong performance across a wide range of experimental scenarios.

**Requested Changes:**

Please address the weakness points.

---

> ### Author Response · Authors · 2026-03-15
> **Response to Reviewer FFXq (1/3)**
>
> We thank you for your thoughtful feedback. We are glad that you find the proposed framework to be simple yet effective, and that the experimental results consistently demonstrate strong performance across LTCIL settings. Your questions on scope and applicability have helped us clarify the positioning of the contribution, and we address each point below.
>
> ## Q1: TH is Not Specific to LTCIL
>
> > [WEAKNESS]: "While the paper frames the head-to-tail transition as a challenge specific to long-tailed class-incremental learning, similar performance degradation due to the shift from full-data access to limited replay buffer has long been observed in standard class-incremental learning settings."
>
> We thank you for highlighting this connection. The observation is correct, and in fact it strengthens the contribution of the paper.
>
> Memory-constrained replay also causes representation degradation in standard CIL. LTCIL does not introduce a completely new problem; rather, it generalizes the standard CIL setting to the more realistic case where class frequencies are unequal. As the imbalance ratio ρ approaches 1, LTCIL converges to standard CIL. The TH phenomenon therefore extends beyond the extreme-imbalance regime. It is a structural consequence of replay-based learning whenever class frequencies differ, and it becomes increasingly pronounced as imbalance grows.
>
> We note that our experiments already evaluate this across multiple imbalance ratios: ρ = 100, 50, and 10 (Tables 1, 2, and 3 in pages 8-9 in the paper). When ρ = 10 (Table 3), the setting is close to balanced and aligns closely with the regime typically studied in standard CIL. In this case, CIL methods are evaluated under conditions that are most favorable to them. Even in this near-balanced setting, DEREK consistently outperforms the strongest CIL baseline (SCoMMER) by large margins: +4.1% on CIFAR 5-task ordered, +4.7% on CIFAR 10-task ordered, +13.9% on ImageNet 5-task ordered, and +13.4% on ImageNet 10-task ordered.
>
> These results suggest that the degradation mechanism you identified in standard CIL is indeed present. The difference is that DEREK addresses this mechanism more effectively, even compared to methods that were originally designed for the balanced case, and does so across the full imbalance spectrum.
>
> ---
>
> ## Q2: Real-World Applicability
>
> > [WEAKNESS]: "The proposed framework relies on the assumption that classes in the current task are data-rich while classes from previous tasks become data-scarce due to fixed-memory replay, leading to an inevitable head-to-tail transition. While this assumption is valid under standard class-incremental benchmarks with standard data sampling strategy for replay buffers, its applicability may be limited in real-world data distributions."
>
> You raise a thoughtful question about real-world applicability, and we are happy to clarify the scope.
>
> DEREK assumes a fixed-size replay buffer with per-class exemplar budgets, which is a standard assumption in the LTCIL literature. We agree that this assumption may not hold in all deployment scenarios, particularly in systems that use dynamic or non-uniform memory allocation. Dynamic allocation strategies such as reservoir sampling are principled alternatives, but they introduce their own trade-offs. Because head classes dominate the training stream, reservoir sampling tends to allocate more buffer slots to them, leaving tail classes with fewer exemplars and sometimes none at all. In practice, this replaces one forgetting problem with another.
>
> At the same time, fixed-budget replay is a natural fit for many real-world constrained deployments. Examples include medical imaging systems where storage is governed by regulatory and infrastructure limits, and satellite or edge devices with fixed onboard memory. These environments also tend to exhibit long-tailed class distributions, which makes the Transitioning Head phenomenon relevant and, by extension, makes our framework applicable rather than an artificial setup.
>
> The one setting where TH does not arise at all is exemplar-free CIL (EFCIL), where no replay buffer is maintained. DEREK is explicitly a replay-based method and does not claim applicability to EFCIL. This is a separate problem regime with its own line of work.
>
> ---

---

> ### Author Response · Authors · 2026-03-15
> **(cont.) Response to Reviewer FFXq (2/3)**
>
> ## Q3: Novelty and Computational Cost
>
> > [WEAKNESS]: "While early knowledge distillation is effective, the core idea of transferring knowledge prior to data scarcity does not appear particularly novel, as it follows naturally from existing knowledge distillation paradigms. In addition, the proposed framework incurs increased computational and memory costs due to full-data knowledge distillation and multi-expert modeling."
>
> We address the two points separately.
>
> **On novelty:** We agree that KD itself is a well-established technique. The contribution of DEREK is not the distillation operation, but the observation that the deterministic transition structure in LTCIL makes proactive KD both possible and necessary, and the method design that follows from this observation. As noted in Section 2 (paragraph 4, page 3) of the revised paper: *"distillation in these methods remains limited, as they rely on small set of exemplars and fail to address TH. Moreover, their tightly coupled head and tail representations hinder early knowledge distillation."* The core difficulty is that three structural questions need to be resolved before early distillation can be applied: which classes will transition, who should serve as the teacher, and when the distillation window opens. Prior methods such as DAKD and SubProto do not have a mechanism to resolve these within their frameworks, which is what prevents them from exploiting the early distillation window. DEREK addresses this by using decoupling to create the teacher–student pair and leveraging the deterministic transition to define the distillation window. These components complement each other by design, and together they make early distillation both well-defined and practically applicable.
>
> We provide two additional pieces of evidence that support this. First, Table Q3-A reports per-group forgetting across ablation conditions, showing that the same Early KD mechanism produces different outcomes depending on whether decoupling precedes it.
>
> |  | Condition | L_head | L_tail | L_CL | Overall Acc ↑ | TH Acc ↑ | Tail Acc ↑ | TH forgetting ↓ | Tail forgetting ↓ | TH−Tail gap ↓ |
> |---|---|---|---|---|---|---|---|---|---|---|
> | A | L_head only (Baseline) | ✓ | ✗ | ✗ | 35.9% | 25.7% | 37.3% | 18.7% | 14.3% | +4.4% |
> | B | Early KD only (Only Stage 2) | ✗ | ✗ | ✓ | 40.0% | 31.6% | 43.8% | 12.2% | 6.9% | +5.3% |
> | C | **DEREK (Stage 1 + Stage 2)** | ✓ | ✓ | ✓ | **44.5%** | **36.1%** | **47.5%** | **8.8%** | 11.8% | **−3.0%** |
>
> **Table Q3-A:** Per-group forgetting across ablation conditions.
>
> At baseline (Row A), TH forgetting (18.7%) is already higher than Tail forgetting (14.3%), a gap of +4.4%. When Early KD is applied without decoupling (Row B), TH forgetting drops to 12.2%, but Tail forgetting drops even more to 6.9%. As a result, the TH–Tail gap actually widens from +4.4% to +5.3%. Without decoupling, Early KD acts as generic distillation and benefits Tail classes more than TH classes. With the full DEREK method (Row C), where decoupling precedes Early KD, the outcome is markedly different: TH forgetting drops to 8.8% while Tail forgetting is 11.8%, flipping the gap to −3.0%. The same distillation mechanism produces opposite effects on the TH–Tail gap depending solely on whether decoupling is present. This indicates that the novelty lies not in the distillation operation itself, but in the structural pairing that DEREK creates before distillation.
>
> Second, Table Q3-B compares Early KD with Late KD under an identical architecture, differing only in distillation timing.
>
> | Method | Overall Acc ↑ | TH Acc ↑ | Tail Acc ↑ |
> |---|---|---|---|
> | Early KD (DEREK) | 44.5% | 36.1% | 47.5% |
> | Late KD | 40.3% | 33.3% | 43.6% |
>
> **Table Q3-B:** Early vs. Late KD under identical architecture, differing only in distillation timing.
>
> Early KD outperforms Late KD on all three metrics: overall accuracy (+4.2%), TH accuracy (+2.8%), and Tail accuracy (+3.9%). The Late KD condition applies distillation at task t+1 using only the available replay exemplars, while the architecture remains identical. The consistent improvement across all groups confirms that the timing enabled by DEREK's design is a meaningful contributor, beyond what can be achieved by applying standard KD after the data loss event.
>
> We have included these tables in the revised paper (Table 5 in Section 5, page 11; Table A10 in Appendix A.2, page 4). We also note that R-uVkU highlighted "Early KD, motivated by the temporal availability of data rather than model architecture alone" as a perspective that may broadly influence knowledge preservation approaches, which we believe captures the contribution well.

---

> ### Author Response · Authors · 2026-03-15
> **(cont. to above) Response to Reviewer FFXq (3/3)**
>
> **On computational and memory cost:** For compute, Table 6 in the revised paper (Section 6, page 12) provides a systematic fairness analysis. As stated in the paper in paragraph 1 of section 6 in pages 11-12: *"With just a single expert (n=1), DEREK already surpasses the state of the art by a healthy margin of +2.7%."* Increasing the capacity of DAKD to 60.2M parameters (2.76×) actually reduces its accuracy from 40.2% to 23.3%. Similarly, tripling the training time does not improve its performance. These results indicate that the gains from DEREK are structural rather than the result of additional capacity or training budget.
>
> Regarding memory cost, the full-data distillation in Stage 2 operates only within task t using the current task's training data. This data is already present in memory for standard training and does not accumulate across tasks. At any point in training, the memory footprint is therefore bounded by the data of a single task, exactly as in standard single-task training. EKD introduces no additional memory overhead beyond what is already required to train on the current task. We have included this clarification in Section 6 (Fairness Analysis, page 12) of the revised paper.

---

### Decision · Action_Editor_e21G · 2026-04-08

**Recommendation:** Accept as is

**Audience:**

Yes

**Audience Explanation:**

Long-tailed recognition community.

**Claims And Evidence:**

Yes

**Claims Explanation:**

This paper identifies Transitioning Heads as a structural source of forgetting in LTCIL and proposes DEREK, which combines representation decoupling with early knowledge distillation to address it. The empirical evaluation is broad and convincing, and the revised manuscript substantially strengthens the claims through targeted additional analyses, including per-group TH evaluation, controlled early-vs-late KD comparison, specialization analysis beyond ensembling, and direct comparison against KL-based distillation. While the core ingredients are individually familiar, the paper’s main contribution is the clear formulation of the TH problem and a well-supported method that addresses it effectively. Overall, the work is technically sound, relevant to the TMLR audience, and meets the standard for acceptance.